# Chaperones rescue the energetic landscape of mutant CFTR at single molecule and in cell

Miklos Bagdany[1], Guido Veit [1], Ryosuke Fukuda[1], Radu G. Avramescu[1], Tsukasa Okiyoneda[1,6], Imad Baaklini[2], Jay Singh[3], Guy Sovak[1,7], Haijin Xu[1], Pirjo M. Apaja[1], Sara Sattin [4], Lenore K. Beitel[1], Ariel Roldan[1], Giorgio Colombo[5], William Balch[3], Jason C. Young [2] & Gergely L. Lukacs[1,2]

Molecular chaperones are pivotal in folding and degradation of the cellular proteome but their impact on the conformational dynamics of near-native membrane proteins with disease relevance remains unknown. Here we report the effect of chaperone activity on the functional conformation of the temperature-sensitive mutant cystic fibrosis channel (ΔF508-CFTR) at the plasma membrane and after reconstitution into phospholipid bilayer. Thermally induced unfolding at 37 °C and concomitant functional inactivation of ΔF508-CFTR are partially suppressed by constitutive activity of Hsc70 and Hsp90 chaperone/co-chaperone at the plasma membrane and post-endoplasmic reticulum compartments in vivo, and at single-molecule level in vitro, indicated by kinetic and thermodynamic remodeling of the mutant gating energetics toward its wild-type counterpart. Thus, molecular chaperones can contribute to functional maintenance of ΔF508-CFTR by reshaping the conformational energetics of its final fold, a mechanism with implication in the regulation of metastable ABC transporters and other plasma membrane proteins activity in health and diseases.

[1] Department of Physiology, McGill University, Montréal, QC, Canada H3G 1Y6. [2] Department of Biochemistry, McGill University, Montréal, QC, Canada H3G 1Y6. [3] Department of Cell and Molecular Biology, Department of Chemistry, The Scripps Research Institute, La Jolla, CA 92037, USA. [4] Università degli Studi di Milano, 20133 Milan, Italy. [5] Istituto di Chimica del Riconoscimento Molecolare, CNR, 20131 Milan, Italy. [6] Present address: Department of Bioscience, School of Science and Technology, Kwansei Gakuin University, 2-1 Gakuen, Sanda 669-1337, Japan. [7] Present address: Anatomy Dep. Canadian Memorial Chiropractic College, Toronto, Canada M2H 3J1. Correspondence and requests for materials should be addressed to G.L.L. (email: gergely.lukacs@mcgill.ca)

Cellular protein homeostasis (proteostasis) networks, including molecular chaperones, have evolved to maintain the functional proteome[1, 2]. Heat shock protein 90 (Hsp90) and 70 (Hsp70/HSPA1A and Hsc70/HSPA8) in concert with co-chaperones facilitate the folding of nascent chains and the refolding of stress-denatured and aggregation prone polypeptides by shielding exposed hydrophobic surfaces, as well as triaging terminally unfolded polypeptides[1]. The Hsp70 family members preferentially recognize unfolded proteins, whereas Hsp90 chaperones bind to partially folded intermediates and account for maintaining the active conformation of their clients[2, 3]. It was proposed that the buffering capacity of chaperones could enhance genetic diversity[4]. In response to proteotoxic stresses, chaperones may mask deleterious changes in the folding energy landscape[1, 5], which can be beneficial in a changing environment.

While it is widely accepted that molecular chaperones assist the folding of soluble proteins by suppressing misfolding and aggregation, evidence that chaperones affect the conformational search by altering the folding landscape energetics is still limited and debated[6]. Although favorable kinetic modulation of the folding pathway by chaperones was recently demonstrated[7, 8], we have no evidence regarding energetic stabilization of the final fold (s) of polypeptides, a central assumption to explain the phenotypic buffering capacity of chaperone networks of marginally stable client proteins[1, 5].

Whereas alterations of secondary structural elements[9] and the folding energy landscape of soluble polypeptides by the cytoplasm have been reported[10], the contribution of molecular chaperones to these processes are not known. The first direct evidence for a favorable impact on the energetic landscape of the denatured FhuA beta-barrel outer membrane protein by the *Escherichia coli* holdase chaperones SurA and Skp was recently reported[11]. Chaperone activity of the Hsp70 homolog, DnaK[12], and the Trigger Factor[8] could also improve the refolding kinetics of soluble model proteins[12], and by analogy, may have a similar effect on the cytosolic regions of multidomain membrane proteins as well.

Preserving plasma membrane (PM) proteostasis is particularly critical for fine-tuning the activity of protein networks responsible for signal transduction, cell–cell communication, ion homeostasis, cell migration, and nutrient uptake. Chaperones have been implicated in the recognition, ubiquitination, internalization, and lysosomal degradation of several conformationally impaired PM proteins that are destabilized by missense mutations, but can partially escape the endoplasmic reticulum (ER) quality control (QC)[13]. It is plausible that molecular chaperones not only contribute to the degradation[13], but also to the conformational maintenance of multidomain membrane proteins in post-ER compartments, representing ~30% of the eukaryotic proteome, considering their complex domain architecture and the intrinsic thermodynamic instability[2, 14].

Cystic fibrosis transmembrane conductance regulator (CFTR) protein is a member of the ATP-binding cassette (ABC) transporter superfamily and comprises two membrane-spanning domains (MSD1 and MSD2) and three cytosolic domains, two nucleotide-binding domains (NBD1 and NBD2) and the unstructured regulatory domain (RD)[15]. We selected the most common cystic fibrosis (CF)-causing mutant, the deletion of the F508 residue (ΔF508) in CFTR, to assess the consequence of molecular chaperon activity on the conformational energetics and function of marginally stable PM proteins that can escape the ER quality control[15].

The cAMP-dependent protein kinase (PKA) stimulated chloride transport activity and density of CFTR variants was monitored as the read-out for the channel functional conformation both at the ensemble level at the PM of airway epithelia and

HeLa cells, as well as at the single-molecule level after the channel reconstitution into black lipid membrane (BLM)[16]. Here, we provide evidence that the cytosolic Hsc70/Hsp90 chaperone systems contribute to the conformational and functional maintenance of the PM resident ΔF508-CFTR at 37 °C. In support, molecular chaperone systems can partially rescue the thermal unfolding-induced functional inactivation of reconstituted ΔF508-CFTR in BLM, reflected by the remodeling of the channel gating energetics towards that of the wild type (WT). These findings demonstrate a mechanistic aspect of the profolding activity of chaperones on a mutant PM protein, with implications in phenotypic modulation of genetic and acquired conformational diseases.

## Results

The ΔF508 mutation, located in the NBD1, disrupts the CFTR domain folding and cooperative domain assembly by destabilizing the NBD1 and the NBD1 and MSD1–2 interfaces[17–20]. The temperature-sensitive folding and stability defect of the ΔF508-CFTR permits its conditional rescue from ER associate degradation at 26 °C and accumulation at the PM[21]. Pharmacological chaperones[22] and proteostasis modulators[23] can also attenuate ΔF508-CFTR misprocessing and misfolding. Regardless of the rescue method, the mature ΔF508-CFTR tends to unfold at post-Golgi compartments at 37 °C, as indicated by its increased protease susceptibility, aggregation propensity, and reduced functional and biochemical half-lives ($t_{1/2} \sim 1–3$ h) as compared to WT ($t_{1/2} \sim 10–12$ h)[16, 24–26]. Intriguingly, both the ΔF508-CFTR folding efficiency at the ER, as well as its stability in post-Golgi compartments and in BLM are increased by the pharmacological chaperone, VX-809[22, 27]. Since the residual chloride secretion via the ΔF508-CFTR that escapes the ER retention correlates with CF disease severity in primary epithelia[28], it is tempting to speculate that this near-native channel population is not only subjected to chaperone-dependent degradation[24], but, perhaps, to refolding as well.

**Chaperones recognize ΔF508-CFTR in post-Golgi compartments**. To exert profolding activity, molecular chaperones should recognize the partially unfolded ΔF508-CFTR in post-Golgi compartments. To test this prediction, complex-glycosylated ΔF508-CFTR (or band C, Fig. 1a) was accumulated at 26 °C for 2 days in stably transfected Baby Hamster Kidney (BHK) cells. Then, to preserve the near-native conformation or unfold the rescued ΔF508-CFTR, cycloheximide (CHX) chase was performed at 26 °C for 10 h or at 37 °C for 2 h, respectively. The CHX-chase ensured the elimination of partially folded core-glycosylated ΔF508- and WT-CFTR from the ER (band B, Fig. 1a), representing folding intermediates prone to chaperone association[19]. Quantitative co-immunopreciptation (Co-IP) showed that unfolding at 37 °C increased the association of complex-glycosylated ΔF508-CFTR with Hsc70, Hsp90 (Fig. 1a, b) and some of their respective co-chaperones, Hdj1 and Aha1 (Fig. 1c)[29, 30], as compared to near-native ΔF508 (26 °C) or WT (37 °C) (Fig. 1b, c). Consistent with ATP-dependent substrate release from Hsc70, ATP depletion of the cell lysate by apyrase augmented the Co-IP of Hsc70 with the complex-glycosylated ΔF508-CFTR (Fig. 1b). These results demonstrate that unfolded ΔF508-CFTR is recognized by the Hsc70/Hsp90 systems in post-Golgi compartments.

**Chaperones increase ΔF508-CFTR biochemical stability**. If Hsc70/Hsp90 activity favorably influences the ΔF508-CFTR conformational stability and function at the PM, chaperone inhibition should promote ΔF508-CFTR unfolding and

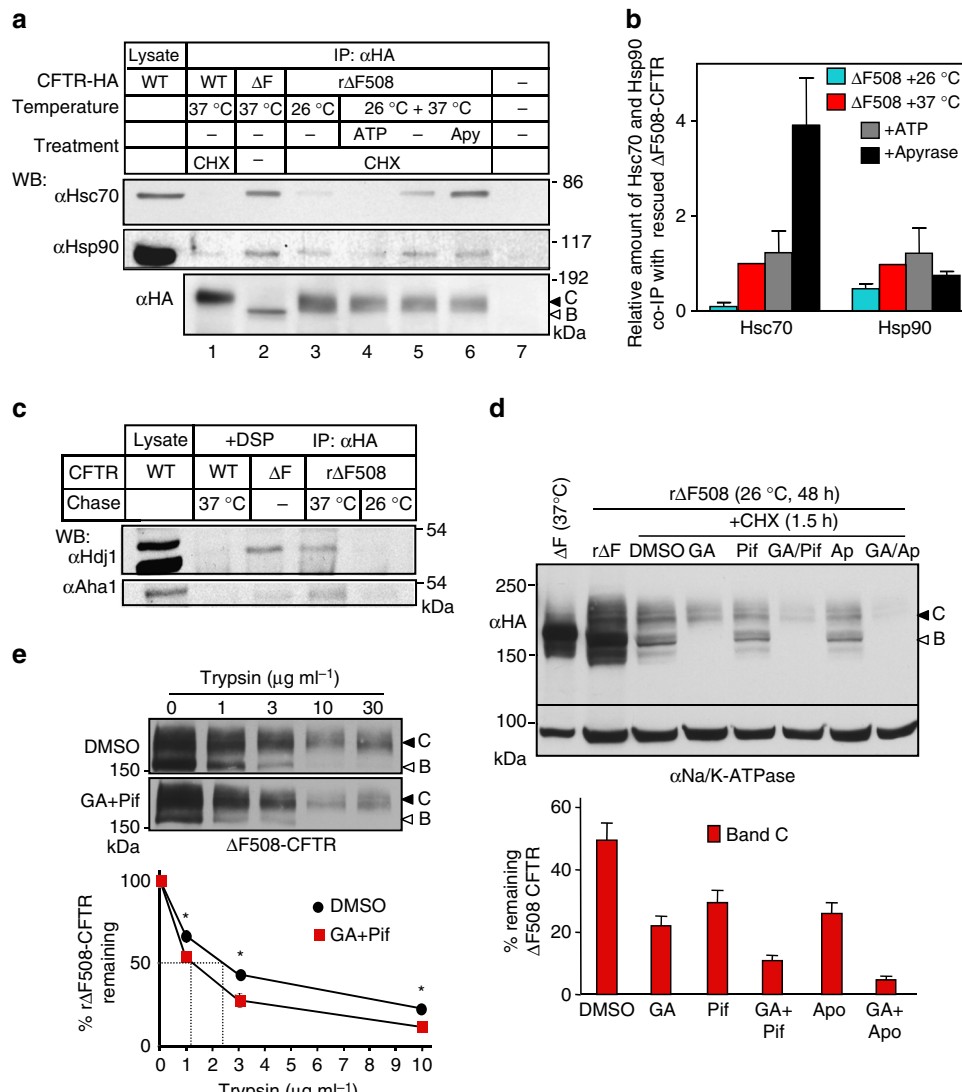

**Fig. 1** Molecular chaperone activity maintains the limited transport competence of ΔF508- and P67L-CFTR. **a** Hsc70 and Hsp90 association with WT- and ΔF508-CFTR containing a C-terminal HA-tag in BHK-21 cells by co-immunoprecipitation (Co-IP) and immunoblotting. Low-temperature rescued ΔF508-CFTR (rΔF508-CFTR, 24 h, 26 °C, *lanes 3–6*), the channel was either unfolded for 2.5 h at 37 °C in the presence of 150 μg ml⁻¹ cycloheximide (CHX) (*lanes 4–6*) or exposed to 37 °C for 20 min and then cultured at 26 °C for 12 h with CHX. The latter protocol preserved the near-native conformation of the complex-glycosylated form (band C, *filled arrowhead*), while ensured degradation of the core-glycosylated form (band B, *empty arrowhead, lane 3*). WT-CFTR cells were exposed to CHX (2.5 h, 37 °C, *lane 1*). CFTR was IP with anti-HA antibody in the absence or presence of 2 mM ATP plus 1 mM MgCl₂ ( + ATP) or with 150 U ml⁻¹ Apyrase ( + Apy) and the IP was probed for Hsc70/Hsp90. Parental (*lane 7*) and non-rescued ΔF508-CFTR (*lane 2*) BHK-21 cells served as controls. **b** Hsc70 or Hsp90 fold-association with the complex-glycosylated rΔF508-CFTR was expressed relative to that observed after unfolding 37 °C for 2.5 h. Data are means ± standard error of the mean (SEM), n = 3–4. **c** Co-IP of co-chaperones with CFTR after crosslinking with 0.1 mM dithiobis[succinimidyl propionate] (DSP) was performed after exposing the cells to CHX chase (2.5 h, at 26 or 37 °C) as described for **a**. **d** The effect of Hsp70/Hsp90 inhibitors on the rΔF508-CFTR turnover, measured by quantitative immunoblotting. Rescued ΔF508-CFTR (26 °C, 48 h) was unfolded for 1.5 h at 37 °C in the absence or presence of Hsc70 (1 μM pifthrin μ [Pif] or 1 μM apoptozole [Apo]) or Hsp90 (10 μg ml⁻¹ GA) inhibitor plus 150 μg ml⁻¹ CHX. Na⁺/K⁺-ATPase served as loading control. Densitometric analysis of rΔF508-CFTR band C remaining after 1.5 h CHX chase. Data are means ± SEM, n = 3–4. **e** Limited trypsinolysis of ΔF508-CFTR in microsomes. Microsomes were isolated from BHK-21 cells that were exposed to DMSO or 5 μM Pif and 5 μg mlᵅ GA (2 h, 37 °C) as described in Methods. Remaining complex-glycosylated rΔF508-CFTR was expressed as the percentage of the initial amount (*bottom panel*). Means ± SEM, n = 3–4

degradation by the peripheral QC in post-Golgi compartments. This was indeed the case. At 37 °C the complex-glycosylated ΔF508-CFTR disappearance was augmented upon inhibition of Hsp90 by geldanamycin (GA) or Hsp70/Hsc70 by apoptozole (Apo, inhibits ATPase turnover) and Pifithrin-μ (Pif, inhibits substrate binding)[31] during a 1.5 h CHX chase, monitored by immunoblotting in BHK-21 cells (Fig. 1d). GA and Pif or Apo additively stimulated the mutant disappearance (Fig. 1d),

suggesting that both chaperone systems contribute to the conformational stabilization of the ΔF508-CFTR in post-Golgi compartments.

Conformational stabilization of the mature ΔF508-CFTR by Hsc70/Hsp90 was assessed by limited trypsinolysis of microsomes, isolated from BHK-21 cells[20]. The protease susceptibility of the complex-glycosylated ΔF508-CFTR was increased by ~twofold after GA + Pif treatment of BHK-21 cells at 37 °C for

2 h, measured by the trypsin concentration required for the 50% degradation of the channel, and probed by immunoblotting (Fig. 1e). This result indicates that Hsc70/Hsp90 activity suppresses the unfolding propensity of the complex-glycosylated ΔF508-CFTR in post-Golgi compartments. Notable, chaperones only partially counteract the mutant unfolding propensity, as the complex-glycosylated ΔF508-CFTR had ~10-fold increased proteases susceptibility at 37 °C as compared to that at 26 °C[24].

**Estimating single CFTR activity by ensemble measurements**. Unfolding of near-native ΔF508-CFTR at the PM is conceivably associated with its functional defect at 37 °C, followed by its ubiquitin-dependent internalization and degradation[24]. The kinetic coupling of these processes is not known. Since the macroscopic transport function of PM CFTR, determined by the channel number, conductance, and open probability, could be targeted by chaperone regulation, both the transport activity and PM density of the mutant were monitored upon modulation of chaperone activity. PKA-stimulated CFTR chloride transport was determined by short circuit current ($I_{sc}$) or halide-sensitive YFP quenching in the human CF bronchial epithelial cell line (CFBE14o- depicted as CFBE) or HeLa cells. The PM density of CFTR was monitored by cell-surface ELISA[24].

We estimated CFTR activation at the single channel level by normalizing the maximal PKA-activated CFTR anion transport with the channel density, a readout designated as Fractional PM Activity (FPMA). The FPMA was validated by demonstrating its increase from ~0.2 to ~0.8–1 upon correction of the ΔF508-CFTR gating defects in the presence of genistein to FPMA similar to that of WT activation in CFBE cells regardless of VX-809 mediated folding correction (Fig. 2a).

**Chaperones delay thermal inactivation of ΔF508-CFTR at PM**. To assess the role of major chaperone systems in the maintenance of CFTR conformation, first they were inhibited by validated siRNA. Hsc70 ablation increased the ΔF508-CFTR PM density by ~threefold in CFBE relative to that in non-targeted (NT) siRNA treated cells (Supplementary Fig. 1a, Fig. 2b). Since ΔF508-CFTR activity remained unaltered, measured with YFP assay (Supplementary Fig. 1b), the relative FPMA was reduced by >60% as compared to NT siRNA exposed cells (Fig. 2b). Similar, but less pronounced changes in the FPMA were observed upon downregulation of Hsp70, Hsp90α, Hsp90β, and some of the co-chaperones tested (Aha1, DNAJA1, and DNAJA2) (Fig. 2b, Supplementary Fig. 1b, c).

Given the influence of major chaperone systems in the channel conformational maturation and ER degradation[32], as well as the profound upregulation of Hsp70, Hsp90α, and Bip after siHsc70 transfection (Supplementary Fig. 1a, d)[24], we switched to short-term (2 h) pharmacological inhibition of chaperones to reduce the indirect effects of siRNA treatments on ΔF508-CFTR folding/stability. Inhibition of Hsc70/Hsp70 or Hsp90 with Pif or GA, respectively, reduced the mutant FPMA in CFBE after 2 h incubation, while had marginal effect on the cellular stress response (Fig. 2c, d). Inhibition of both Hsc70 and Hsp90 acted additively and led to 54 ± 6% decrease of the mutant FMPA as compared to WT-CFTR (20 ± 4%, $p < 0.01$) (Fig. 2c, d). Similar results were obtained by using the structurally distinct Hsc70/Hsp70 (Apo and MKT-077 [MKT], an allosteric inhibitor of ATP turnover or Hsp90 (Genitispib [Gpib]) alone or in combination (Supplementary Fig. 1e, f), supporting the contribution of both Hsp90 and Hsp70/Hsc70 systems to the conformational maintenance of ΔF508-CFTR at the PM and in post-ER compartments in CFBE. Similar changes were observed in HeLa cells (Supplementary Fig. 1g).

The engagement of molecular chaperones in the conformational rescue of P67L-CFTR, another CF-causing processing mutant, was also evaluated. Pharmacological inhibition of Hsp90 + Hsc70 (37 °C, 2 h) reduced the FPMA of P67L significantly more than the WT (35.7 ± 5 vs. 20 ± 4%, $p < 0.05$) (Fig. 2d, e). This result indicates that the conformational maintenance activity of chaperones is not restricted to ΔF508-CFTR.

**Modulation of ΔF508-CFTR chaperone-sensitivity at PM**. If the Hsc70/Hsp90 activity partially suppresses the intrinsic instability of the PM ΔF508-CFTR, genetic or pharmacological conformational stabilization should render the mutant resistant to chaperone inhibition and vice versa. Indeed, suppressor mutations (3S and/or R1070W, defined in Supplementary Table 1) largely preserved the FPMA of ΔF508-CFTR at the WT level upon chaperone inhibition (Fig. 2f), consistent with the increased protease resistance of ΔF508-CFTR-3S-R1070W relative to ΔF508[20]. Remarkable, comparable chaperone-resistance was documented after VX-809 treatment, an FDA approved pharmacological chaperone that binds to and stabilizes the ΔF508-CFTR[22, 27, 33] (Fig. 2g). VX-809 also delayed the mutant channel accelerated lysosomal delivery in the presence of GA + Pif (Supplementary Fig. 2a, b). Conversely, destabilizing ΔF508-CFTR at the ER and PM by exposing the CFBE to the potentiator drug, ivacaftor (VX-770)[34], enhanced the contribution of chaperones to the conformational maintenance, as revealed by the augmented downregulation of ΔF508-CFTR-3S upon chaperone inhibition (Supplementary Fig. 2c, d). These results, collectively, suggest that the Hsc70/Hsp90 chaperone network partially counteracts the conformational and biochemical destabilization of ΔF508-CFTR in post-Golgi compartments, including the PM and endosomes.

**Thermal inactivation of ΔF508-CFTR in black lipid membrane**. To conclusively demonstrate the impact of recombinant chaperone activity on the near-native ΔF508-CFTR unfolding, first we reconstituted the mutant into black lipid membrane (BLM) to observe its inactivation process. Following in vitro phosphorylation, CFTR containing microsomes[20] were fused to the BLM in a thermostated chamber[16] at ~24 °C. This technique ensured sustained phosphorylation of CFTR after addition of the PKA catalytic subunit and in the absence of endogenous phosphatases (Supplementary Fig. 3a)[35]. Furthermore, microsome reconstitution rendered the BLM virtually free of cytosolic chaperones and permitted monitoring the mutant activity without proteolytic degradation (Supplementary Fig. 3b).

CFTR gating was routinely monitored upon increasing the temperature from 23 to 36 °C at ~1.4 °C min$^{-1}$ rate at −60 mV holding potential. The open probability ($P_o$) and the conductance of WT-CFTR were increased from ~0.3 to ~0.45 and 9.6 ± 0.3 pS to 12.3 ± 0.4 pS, respectively (Fig. 3a, f, Supplementary Fig. 3c, d), as a signature of the WT thermal activation, as reported[16, 36]. In contrast, the initial activity of rescued ΔF508-CFTR ($P_o$~0.1) and ΔF508-CFTR-2RK variant ($P_o$~0.2), containing the R29K and R555K second site mutations, was reduced by ~50% at 36 °C with an inactivation half-life ($t_{1/2}$) of ~4 and ~8 min, respectively (Fig. 3b, c, f, Supplementary Fig. 3e, and Supplementary Table 1). The 2RK mutation enhanced the recording success of the ΔF508-CFTR without masking its processing defect and thermal instability[16, 27] and was preferentially used in subsequent studies.

Remarkably, the inactivation half-life ($t_{1/2}$) of rescued ΔF508-CFTR and ΔF508-CFTR-2RK, measured by the $P_o$ decay kinetics, was ~15-fold faster in the BLM at 32–36 °C than at the PM ($t_{1/2}$~120 min, 37 °C) of CFBE or HeLa cells, but was comparable to that in excised patches[16, 34, 37–39] at 37 °C ($t_{1/2}$~3–8 min)

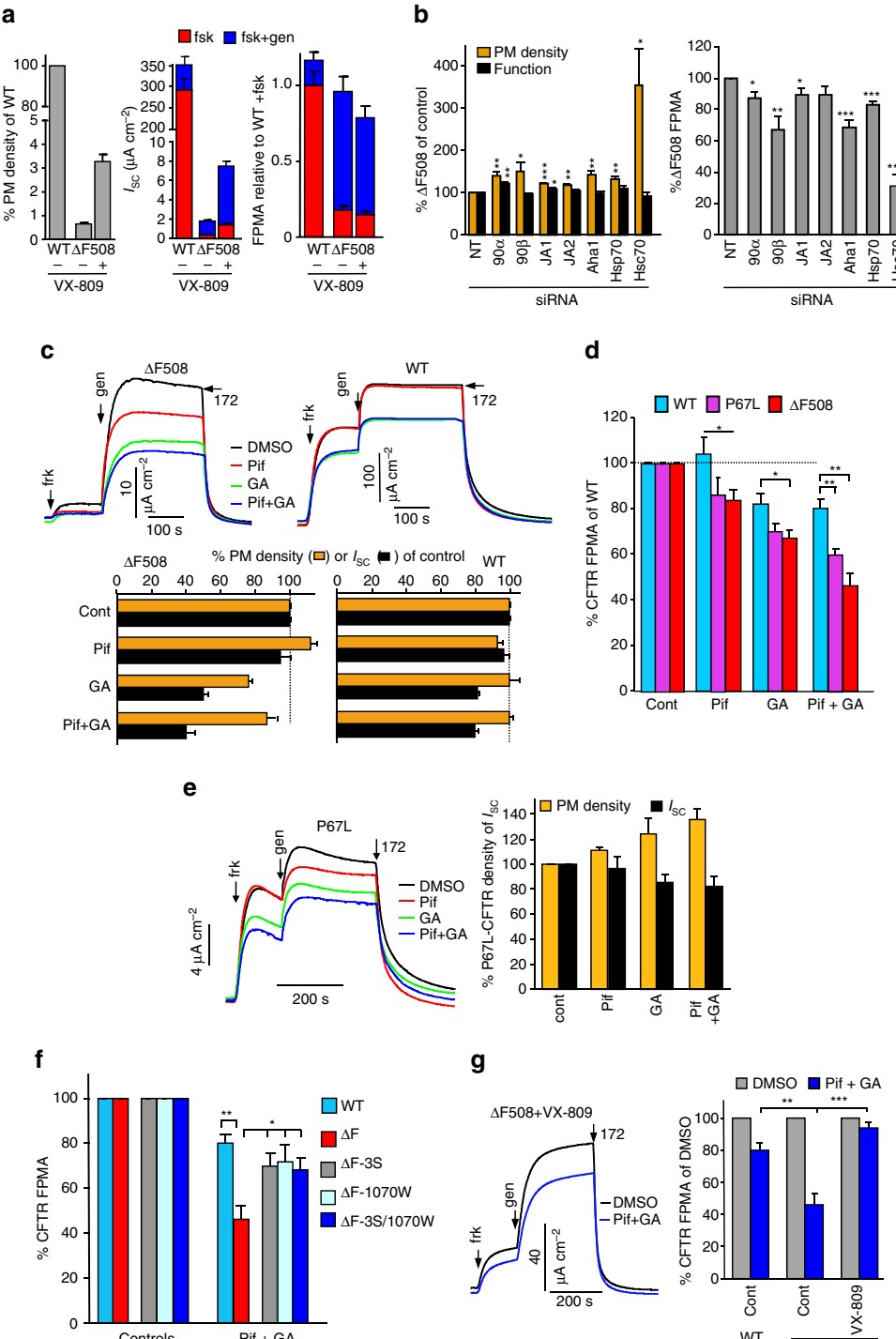

**Fig. 2** Molecular chaperone activity maintains the limited transport competence of ΔF508- and P67L-CFTR. **a** FPMA of WT and ΔF508-CFTR calculated from PM density and short-circuit current ($I_{sc}$) with and without 24 h 3 μM VX-809 rescue, expressed as percentage of WT-CFTR expressing CFBE (*left panel*). $I_{sc}$ was measured after sequential addition of 20 μM forskolin (fsk) and 100 μM genistein (gen), followed by CFTR inhibition with Inh$_{172}$ (172, 20 μM). The FPMA (*right panel*) was calculated as the ratio of PM density and $I_{sc}$, and normalized to WT. Means ± SEM, $n = 3$; *error bars* in the *right panel* are error propagation. **b** The effect of chaperone or co-chaperone knockdown on PM density, function and FPMA of the ΔF508-3HA in CFBE. The rescued ΔF508-CFTR (30 °C, 48 h) was unfolded for 2 h at 37 °C. Hsp90α (90α), Hsp90β (90β), DNAJA1 (JA1), DNAJA2 (JA2), Aha1, Hsp70, and Hsc70 were silenced with 50 nM siRNA. Non-targeted siRNA served as controls. Function was determined by YFP quenching assay. **c** The effect of Hsc70 and Hsp90 inhibition on the PM density and function of the ΔF508- and WT-CFTR-3HA in CFBE. The rescued ΔF508-CFTR (30 °C, 48 h) was unfolded (2 h at 37 °C) in the presence of 5 μM Pif and/or 5 μg ml$^{-1}$ GA. $I_{sc}$ was measured as in **a**. Representative measurements are in the *top panels*. **d** The effect of Hsc70 and Hsp90 activity on the FPMA of the ΔF508- and P67L-CFTR in CFBE cells. Experiments were performed as in **c**. **e** $I_{sc}$ and PM density of P67L-CFTR in CFBE as described in **c**. **f** The effect of Hsc70 and Hsp90 activity on ΔF508-CFTR FPMA in CFBE cells. CFBE cells, expressing ΔF508-CFTR with second site mutations or rescued ΔF508-CFTR (30 °C, 48 h), were treated for 2 h at 37 °C with 5 μM Pif and 5 μg ml$^{-1}$ GA. Experiments were performed as in **c**. **g** Conformational rescue of the ΔF508-CFTR by low temperature plus VX-809 (3 μM, 24 h) prevents the loss of ΔF508-CFTR FPMA by Pif + GA in CFBE cells. Experiments were performed as in **c**. Data are means ± SEM, $n = 3$–4. *$P < 0.05$, **$P < 0.01$ and ***$P < 0.001$

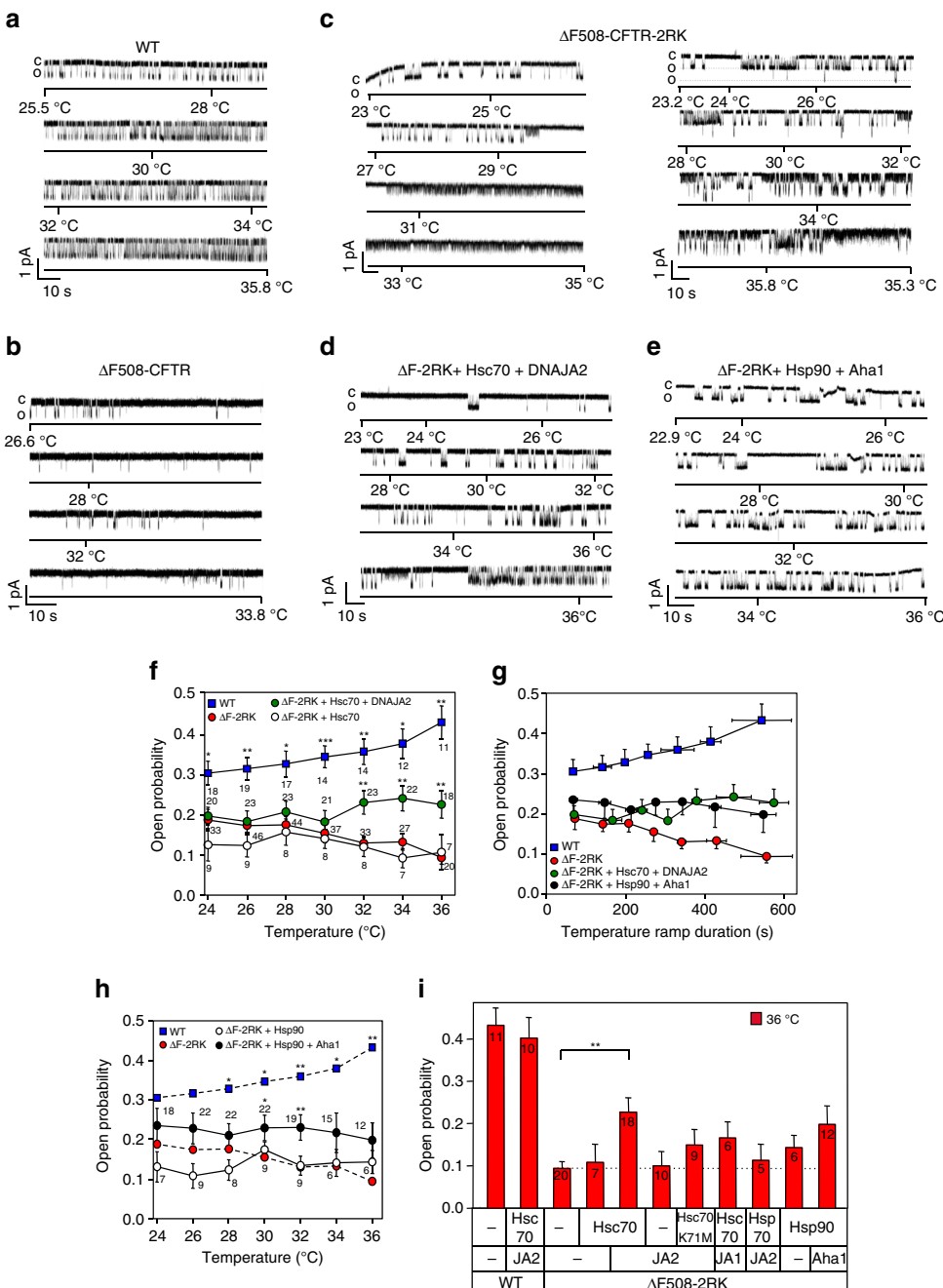

**Fig. 3** Molecular chaperones partially revert thermal inactivation of ΔF508-CFTR in black lipid membrane (BLM). **a–e** Single-channel records of WT-, ΔF508- and ΔF508-CFTR-2RK channel temperature-dependent activity in the absence **a–c** or presence of Hsc70 (2 μM) and DNAJA2 (2 μM) **d** or Hsp90 (2 μM) and Aha1 (2 μM) **e**. The predominantly observable thermal inactivation of ΔF508-CFTR-2RK is shown on the *left* of **c**. Phosphorylated channels were reconstituted and recorded in BLM as described in Methods. Closed (c) and open (o) states are indicated. Two channels were incorporated into the BLM in the *right panel* of **c**. **f** The effect of Hsc70 or Hsc70 + DNAJA2 on the temperature-dependent open probability ($P_o$) of the ΔF508-CFTR-2RK determined in 2 °C intervals as in **a–d**. **g** Time-dependent changes in the $P_o$ of WT- and ΔF508-CFTR-2RK during temperature ramps in the presence of the indicated chaperones. **h** Hsp90/Aha1 (2 μM), but not the Hsp90 alone, confers resistance against thermal inactivation of ΔF508-CFTR-2RK. **i** Effect of molecular chaperones and co-chaperones on the mean $P_o$ of WT- and ΔF508-CFTR-2RK at 36 °C. The concentration of Hsp70, Hsc70-K71M and DNAJA1 (JA1) was 2 μM. Data are means ± SEM, *$P < 0.05$, **$P < 0.01$ or ***$P < 0.001$. $P_o$ values of ΔF508-CFTR-2RK in the absence of chaperones have been derived from data in Veit et al.[34] Number of independent records for channel activity is indicated on **f** and **h**

(Fig. 3g, Supplementary Fig. 3f). Since inactivation of PKA can be ruled out based on the increasing $P_o$ of WT-CFTR during temperature ramps (Supplementary Fig. 3a)[35], the accelerated inactivation may be attributed to the absence of molecular chaperone profolding activity in the BLM, a presumption tested next.

**Chaperones favorably remodel ΔF508-CFTR gating energetics.** Recombinant Hsc70 with DNAJA2[40] that activates the Hsc70 ATPase, partially reversed the mutant thermal inactivation of ΔF508-CFTR-2RK, as indicated by the increased $P_o$ at >30 °C. In the presence of Hsc70/DNAJA2 the $P_o$ increased by ~2-fold at 36 °C (Fig. 3d, f, i). Similarly, Hsp90 in combination with Aha1

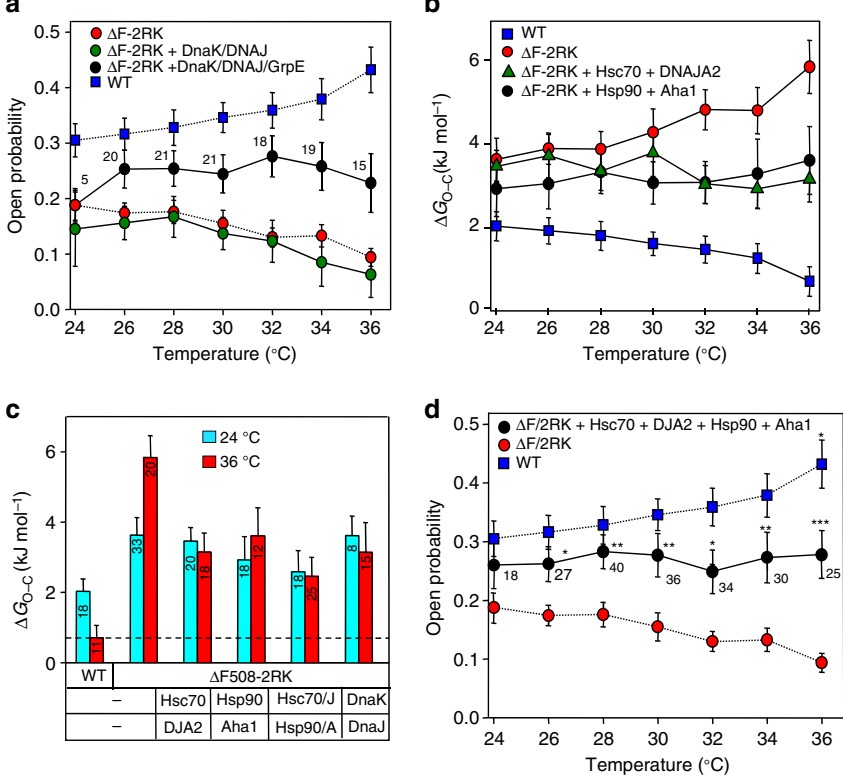

**Fig. 4** Molecular chaperones shift the open and closed state folding energetics of ΔF508-CFTR towards that of the WT during thermal unfolding.
**a** Temperature-dependent single-channel activity of rΔF508-CFTR-2RK in the presence of DnaK (1 µM), DnaJ (0.2 µM), and GrpE (0.5 µM) as indicated. $n = 5$–21. **b** The equilibrium steady-state Gibbs free energy difference ($\Delta G_{O-C}$) between the open (O) and closed (C) states of WT and ΔF508-2RK was calculated in the presence or absence of Hsc70/DNAJA2 or Hsp90/Aha1 based on the $\Delta G_{O-C} = -RT(\ln K_e)$ equation. **c** The $\Delta G_{O-C}$ values of WT and ΔF508-CFTR at 24 and 36 °C are depicted from **a** (DnaK/DnaJ/GrpE), **b** (Hsp90/Aha1, Hsc70/DNAJA2) and **d** (Hsp90/Aha1 + Hsc70/DJA2).
**d** Temperature-dependent $P_o$ of ΔF508-CFTR-2RK in the presence of Hsp90/Aha1 plus Hsc70/DNAJA2 (DJA2) as in Fig. 3a–f. Means ± SEM, $n$ values **b**–**d** are the same as indicated for the relevant $P_o$ in Fig. 3f and h. *$P < 0.05$, **$P < 0.01$ or ***$P < 0.001$

that activates the chaperone ATPase[2], conferred partial thermal-resistance to ΔF508-CFTR-2RK activity and increased the $P_o$ by ~1.5–2.1-fold at 30–36 °C (Fig. 3e, h, i).

The following observations indicate that specific chaperone–cochaperone interactions and the ATPase cycle activation are required for reverting the mutant gating defect[41]. In the absence of the co-chaperones DNAJA2 and Aha1, the conformational stabilization of the mutant by Hsc70 and Hsp90, respectively, was reduced (Fig. 3f, h, i and Supplementary Fig. 4a–f). Likewise, the ΔF508-CFTR-2RK thermal inactivation was not attenuated by the ATPase deficient Hsc70-K71M variant[40], replacing DNAJA2 with DNAJA1 or by using DNAJA2 alone (Fig. 3i, Supplementary Fig. 4a–f). Notably, the gating of the native WT-CFTR was insensitive to Hsc70/DNAJA2 activity (Supplementary Fig. 3h).

To reinforce the role of ATPase activity of Hsc70 in stabilizing the near-native mutant, we took the advantage of the extensively characterized DnaK/DnaJ/GrpE prokaryotic chaperone system[42], which has a similar peptide client recognition profile to Hsc70[43] and its refolding activity is stimulated by the nucleotide exchange factor GrpE (Supplementary Fig. 4h)[44]. DnaK/DnaJ/GroE jointly elicited comparable thermal stabilization of ΔF508-CFTR open state as Hsc70/DNAJA2 (Fig. 4a, Supplementary Fig. 4i, j). However, the stabilizing effect was abolished in the absence of GrpE + DnaJ or GrpE, indicating that accelerated nucleotide exchange is indispensable for the mutant refolding (Fig. 4a).

The Gibbs free energy difference between the open (O) and closed (C) state of the channel (designated as $\Delta G_{O-C}$) was calculated[45], assuming a simplified, two-state gating model at

near equilibrium during the slow temperature ramp[45, 46]. The WT O state thermodynamic stability relative to the C state was considerably higher ($\Delta G_{O-C}$~0.8 ± 0.4 kJ mol$^{-1}$) than for the mutants (ΔF508-CFTR-2RK: $\Delta G_{O-C}$~5.8 ± 0.6 kJ mol$^{-1}$ and ΔF508-CFTR: $\Delta G_{O-C}$~8.3 ± 0.6 kJ mol$^{-1}$), impeding ΔF508 open state prevalence at 36 °C (Supplementary Fig. 4g, *right panel*). Both Hsc70/DNAJA2 and Hsp90/Aha1 reduced the $\Delta G_{O-C}$ of ΔF508-CFTR-2RK by ~50% at 36 °C (Fig. 4b, c). Comparable stabilization was achieved by DnaK/DnaJ/GroE chaperones (Fig. 4a, c). Combination of Hsc70/Hsp90 chaperone systems further improved the open state stability ($\Delta G_{O-C} = 2.5 ± 0.5$ kJ mol$^{-1}$) (Fig. 4c, d).

**Hsc70 improves ΔF508 gating kinetics and thermodynamics.** CFTR gating kinetics is defined by the opening ($\Delta G^{\ddagger}_{C-O}$) and closing ($\Delta G^{\ddagger}_{O-C}$) activation energies and calculated from dwell time histograms of single channel O and C states. The WT mean closed time ($\tau_C$) decreased from ~173 to ~114 ms, while the mutant $\tau_C$ increased from ~134 to ~700 ms between 30 and 36 °C (Fig. 5a, Supplementary Fig. 5a). The WT openings had a single O state with a mean open time ($\tau_O$) of ~93–160 ms. The ΔF508 displayed a bimodal $\tau_O$ distribution, with a short ($\tau_{O1}$ ~10–30 ms, O1) and a long ($\tau_{O2}$ ~352–570 ms, O2) O states between 24 and 36 °C (Fig. 5b, *left and middle panels*), consistent with a three-state gating model (Supplementary Fig. 5c). The contribution of the short O1 state to the mutant activity was increased, while the longer O2 state was decreased at 32–36 °C (Fig. 5b, Supplementary Table 2). The mean $\tau_{O2}$ was also reduced by ~30%, further

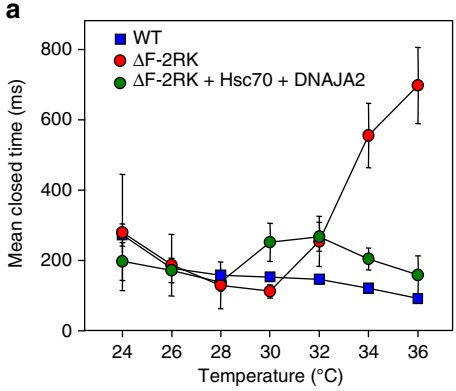

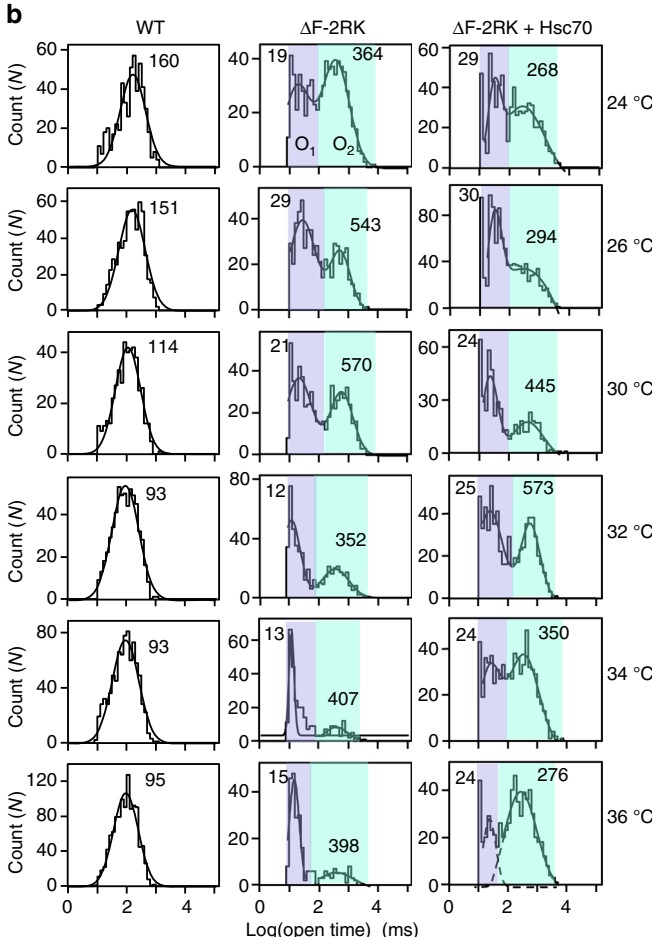

**Fig. 5** Hsc70/DNAJA2 shifts the gating kinetics of single ΔF508-CFTR towards that of the WT during thermal unfolding. **a** The temperature dependence of mean closed times ($\tau_C$) of WT, ΔF508-2RK and ΔF508-2RK with Hsc70/DNAJA2 was determined from dwell time histograms restricted to single channel records of ΔF508-2RK (total recording time was 10–28 min, n = 5–24), ΔF508-2RK with Hsc70/DNAJA2 (15–25 min, n = 10–23) and WT (4–7 min, n = 3–7). Data are means ± SEM. **b** Open dwell time histograms of single WT, ΔF508-2RK and ΔF508/2RK channels with Hsc70/DNAJA in the BLM. Histograms were fitted with one or two components Gaussian distribution. The mean O1 and O2 times are indicated in ms

decreasing the $P_o$ of ΔF508 at 36 °C (Fig. 5b, Supplementary Fig. 5b and Supplementary Table 2).

Remarkably, Hsc70/DNAJA2 activity decreased the ΔF508 mean $\tau_C$ by ~4-fold (from ~700 to ~180 ms) and retained the

channel in the longer O2 state at 32–36 °C (Fig. 5b, Supplementary Fig. 5b and Supplementary Table 2). Since transitions between the O1 and O2 states were extremely rare (Supplementary Table 3), the energetic calculation was based on the $O1_{(T)}-C_{(T)}-O2_{(T)}$ gating model, where the C state likely includes more than one conformation and all states are susceptible to unfolding and conformational remodeling by chaperones at >31 °C. By calculating the $\Delta G_{O-C}$ from the opening and closing activation energies ($\Delta G^{\ddagger}_{C-O}$ and $\Delta G^{\ddagger}_{O-C}$), we conclude that Hsc70/DNAJA2 decreased both the kinetic and thermodynamic barrier of the mutant opening at 36 °C (Fig. 6a, b). The active state stabilization could be attributed to decreased entropic energy requirement that was partially off-set by increased enthalpy of the open states by chaperone (Supplementary Fig. 5e). The $O1_{36\,°C}$ and $O2_{36\,°C}$ were stabilized by $\Delta\Delta G_{O-C}$ ~4.8 and ~3.2 kJ mol$^{-1}$, respectively, by Hsc70 (Fig. 6a, b, Supplementary Table 4). Only a modest effect was observed on the mutant at 24 °C and on the WT (24–36 °C), consistent with their near-native and native conformation, respectively (Supplementary Table 4, Supplementary Fig. 5f). Thus, molecular chaperones can partially buffer the severity of mutant conformational and functional defect at 36 °C by shifting the gating energy landscape towards the WT (Fig. 6b).

**Molecular chaperones target domain–domain assembly defects.** If chaperones recognize the non-native cytosolic domains of ΔF508-CFTR, conformational stabilization of the NBD1 and NBD1–MSD1/2 interface by second site suppressor mutations[18, 47, 48] should render the mutant less susceptible to chaperone activity in the BLM, as we observed in CFBE (Fig. 2f). Accordingly, correction of the NBD1–MSD1/2 interface and NBD1 folding defects by the R1070W and 3S suppressor mutations or R1070W[18] and low temperature rescue[20], respectively, were sufficient to restore the $P_o$ and the $\Delta G_{O-C}$ of ΔF508-CFTR close to that of WT at 36 °C (Fig. 7a, b). The open state of ΔF508-CFTR-R1070W became insensitive for stabilization by Hsc70/DNAJA2 similar to that of the WT (Fig. 7a, b, Supplementary Fig. 7a–d).

Stabilizing the ΔF508-NBD1 by the 3S mutation (Supplementary Fig. 3g) prevented the functional and gating energetic correction by Hsc70/DNAJA2 (Fig. 7a, b, Supplementary Fig. 6e, f). This suggests that stabilization of ΔF508-NBD1 folding defect alone may reduce chaperone binding and/or uncouple the channel cooperative domain assembly, which is completed post-translationally with coupled domain folding and chaperone assistance in vivo[19, 20, 29, 49, 50]. This inference is supported by the observations that promoting CFTR coupled domain folding by NBD1–NBD2 dimerization stabilized the ΔF508-CFTR against thermal inactivation in the BLM. The NBDs dimerization was facilitated by: (1) introduction the R1S revertant and suppressor mutations[51] (Supplementary Fig. 3g, Supplementary Table 1), (2) inhibiting the ATPase activity of the composite site 2 of NBD dimer by the E1371S mutation[52], or (3) using the hydrolysable ATP analog, 2′deoxy-ATP (dATP)[53]. The NBD1–NBD2 dimer stabilization was reflected by the increased $P_o$ and reduced $\Delta G_{O-C}$ of the mutant at 36 °C (Fig. 7c, d). Jointly, these results are consistent with the notion that the chaperone-mediated conformational stabilization of ΔF508-CFTR can be mimicked by enhanced NBD1–NBD2 heterodimerization.

**Chaperones activation enhances ΔF508-CFTR PM function.** If the folding capacity of the proteostasis network is limited in CF epithelia, increasing the chaperone activity may improve ΔF508-CFTR function and stability at the PM. To test this hypothesis, we first augmented the stress-independent activation

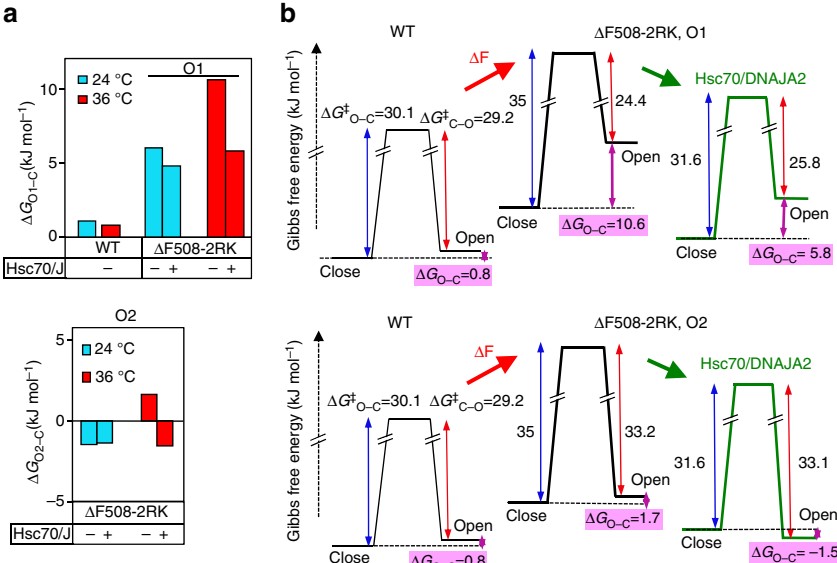

**Fig. 6** Molecular chaperones reshape the transition and open state gating energetics of ΔF508-CFTR during thermal unfolding. **a, b** Equilibrium Gibbs free energy differences ($\Delta G_{O-C}$) between the open (O1 and O2) and closed (C) states were calculated from the difference of the opening and closing activation energies ($\Delta G_{O-C} = \Delta G^{\ddagger}_{O-C} - \Delta G^{\ddagger}_{C-O}$, Supplementary Table 4). $\Delta G^{\ddagger}_{O-C}$ and $\Delta G^{\ddagger}_{C-O}$ were determined from the gating kinetics of single channels at 36 °C as described in Methods. **b** Schematic representation of the gating energetics of WT and ΔF508-CFTR-2RK in the absence or presence of Hsc70/DNAJA2 at 36 °C based on calculation described in **a**. Hsc70/DNAJA2 predominantly decreases the opening activation energy of the ΔF508-2RK. The closed state folding free energy of ΔF508-CFTR-2RK after thermal unfolding relative to WT is arbitrarily chosen. The $\Delta G_{O-C}$ values are derived from **a**

of the heat shock response by co-expressing the constitutively active heat-shock factor-1 (cHSF1) fused to the destabilized mutant FK506 binding protein 12 (FKBP-cHSF1)[54] in ΔF508-CFTR HeLa cells. Degradation of the FKBP-cHSF1 fusion was prevented by the small molecule Shield-1, a stabilizer of FKBP that causes transcriptional activation of the HSR with over-expression of several cytosolic components of the ATP-dependent chaperoning pathway (e.g., Hsp90, Hsp70s, and Aha1) and small heat shock proteins as described[54] (e.g., Fig. 8a, Supplementary Fig. 8a). cHSF1 expression slowed down the ΔF508-CFTR constitutive and GA-induced turnover at the PM (Fig. 8b). Furthermore, cHSF1 augmented the FPMA of ΔF508-CFTR, which could be attributed to improved channel function despite reduced PM density of the mutant (Fig. 8c).

To assess the impact of the acutely increased Hsp90 activity on ΔF508-CFTR functional and biochemical turnover at the PM, we used the recently developed 2-phenyl benzofurane derivates (CheCOSP-26, -27, -30 and -36), which were designed to allosterically activate the Hsp90 ATPase cycle by binding to the boundary between the middle and C-terminal domain of Hsp90[55, 56]. These drugs acutely counteracted the GA-induced ErbB2 destabilization at the PM (Supplementary Fig. 8b). Exposing CFBE cells to CheCOSP-26, -27 or -30 for 2 h at 37 °C, augmented the ΔF508-CFTR PM function, measured by the YFP assay, while modestly decreased its PM density. This caused a ~1.6–2-fold increase in the FPMA, which was sensitive to the CFTR blocker, Inh$_{172}$ (Fig. 8d). The CheCOSP compounds also delayed the mutant inactivation after 2 h at 37 °C, as indicated by the 3-fold increased FPMA, compared to non-treated cells, a phenomenon that prevailed in the presence of VX-809 (Fig. 8e). These are consistent with the notion that stimulation of Hsp90 cytosolic profolding activity can favorably change the function and stability of ΔF508-CFTR at the PM.

## Discussion

Although Anfinsen's dogma postulates that the native state is determined by the protein's amino acid sequence, accumulating evidences suggest that the polypeptide folding and its final outcome may be influenced by the nascent chain elongation rate, interaction with the ribosome, posttranslational modifications, chaperone activity as well as cytosolic crowding and composition[2, 9, 57–59]. Jointly, these were referred to as the physiologic state by Anfinsen[60]. While chaperone-induced suppression of misfolding and aggregation to increase the folding yield of soluble proteins have been established, the malleability of the final fold and folding pathway by chaperones is not fully understood. One of our most important findings is that the activity of Hsc70/DNAJA2, Hsp90/Aha1 or DnaK/DnaJ/GrpE chaperone systems can shift a mutant PM channel conformation towards the native fold by reshaping the kinetic and thermodynamic determinants of the gating energetics at the single-molecule level. Furthermore, we show that the molecular chaperones, key players of the cellular proteostasis network, are not only involved in the degradation[13] but can also suppress the functional folding defects of mutant PM proteins by influencing the "final" fold of polypeptides.

The intracellular environment can cause structural changes to intrinsically unstructured regions of the yeast prion protein Sup35, as compared to its isolated native form, measured by sensitivity-enhanced NMR[9]. The cytosol can also influence the rate of folding and the thermodynamic stability of phosphoglycerate kinase (PGK), detected by Forster resonance energy transfer[61]. These experiments did not identify constituents of the cytosolic proteostasis network that are responsible for structural changes of Sup35 and PGK. More recent results at single-molecule level, however, suggested that both holdase (Skp) and foldase (DnaK/DnaJ) chaperones can reduce the activation energy of a membrane and a soluble protein (re)folding, respectively[11, 12]. Similar conclusions were reached by examining the effect of the trigger factor and GroEL/GroES on the (re)folding kinetics of soluble client proteins at single molecule and ensemble level, respectively[7, 8], without assessing the final fold energetics. Our results provide evidence for both thermodynamic and kinetic stabilization of a mutant integral PM protein

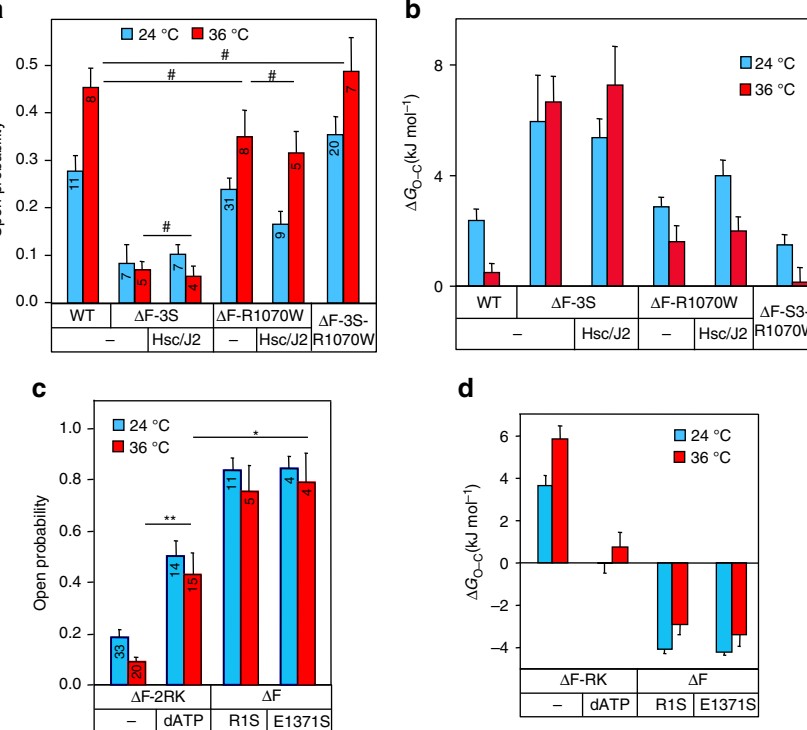

**Fig. 7** Chaperoning of $\Delta$F508-CFTR unfolding is altered by second site mutations in the BLM. **a, b** The influence of second site suppressor mutations (3S, R1070W and 3S + R1070W) and Hsc70/DNAJA2 (Hsc/J2) activity on the $P_o$ **a** and $\Delta G_{O-C}$ **b** of the $\Delta$F508-CFTR-3HA at 24 and 36 °C in BLM. The $P_o$ and $\Delta G_O$ were measured as in Figs. 3 and 4. **c, d** Stabilization of NBD1 and/or NBD1–NBD2 dimer was accomplished by R1S, E1371S mutations, or by the inclusion of 2′-deoxyadenosine 5′-triphosphate (dATP, 2 mM). The Gibbs free energy of opening was calculated based on the $\Delta G_{O-C} = -RT(\ln K_e)$ equation. Representative records are shown in Supplementary Fig. 7a, b, d–f. The reconstituted channels were characterized as in Figs. 2 and 4. Data are means ± SEM, $n$ is defined in **c**. $^\#P \geq 0.05$, $^*P < 0.05$ or $^{**}P < 0.01$. The $P_o$ values of R1S and E1371S mutants were derived from Veit et al.[34]

($\Delta$F508-CFTR) by molecular chaperones. This mechanism may modulate the phenotypic manifestation of conformational diseases, caused by mutations in the ABC transporter superfamily[62].

The recruitment of the Hsc70 and Hsp90 chaperon–cochaperone systems to the complex-glycosylated $\Delta$F508-CFTR during thermal unfolding (Fig. 1a–c) forms the biochemical basis for reshaping the mutant unfolding trajectory both at the PM and in BLM. This inference is supported by the inverse correlation observed between the Hsc70/Hsp90 chaperone activity and the severity of the mutant conformational and functional defect. Attenuated or enhanced folding activity of cytosolic chaperones led to decreased or increased function/stability of $\Delta$F508-CFTR, respectively, at the PM. Both the ATPase cycle of Hsc70 and the presence of DNAJA2, but not DNAJA1, are required for the open state stabilization of the PM $\Delta$F508-CFTR at 36 °C. Likewise, activation of Hsp90 or DnaK/DnaJ ATPase cycle by Aha1 or GrpE, respectively, was required to increase the mutant open state stability, implying that beyond binding, the chaperone ATPase cycle is indispensable for the mutant conformational rearrangement. Considering that $\Delta$F508-CFTR largely resides in closed state during the manipulation of molecular chaperone activity in the absence of exogenous PKA activator, we suggest that both open and closed states are susceptible to conformational stabilization by chaperones.

While we still lack detailed molecular understanding of chaperones profolding activity, it is conceivable that they protect unfolding intermediates from further intramolecular misfolding and thereby prevent coupled domain disassembly of CFTR[50]. In addition, entropic destabilization and stabilization of the mutant during association–dissociation cycle of chaperones may permit to relaunch the conformational search of unfolded cytosolic

domains for WT-like folding pathway(s). Molecular chaperones may exert similar influence on the co- and post-translational folding energy landscape of WT CFTR, considering that the conformation of early folding intermediates of WT- and $\Delta$F508-CFTR partially overlap[49, 63]. In support, the limited folding efficiency of WT-CFTR is enhanced by the NBD1 and the NBD1–MSD interface stabilization[20] and severely reduced by inhibiting Hsp90 activity[64].

The preferential requirement of DNAJA2 for folding over DNAJA1, which contributes to CHIP- and ubiquitin-dependent degradation of unfolded complex-glycosylated $\Delta$F508-CFTR from the PM, provides additional support for the specialized role of distinct chaperone–co-chaperone complexes in a variety of cellular processes, involving conformationally perturbed client proteins[65, 66]. Importantly, the gating thermodynamics of the $\Delta$F508-CFTR during thermal unfolding was similarly altered by the activity of Hsp90/Aha1 and DnaK/DnaJ/GrpE chaperone system in the BLM. These observations, jointly, demonstrate that the proteostasis system can favorably influence both the kinetic barrier to refolding, and the relative thermodynamics of the final state, and suggest that the final conformation(s) of the mutant is influenced by molecular chaperones.

The more pronounced stabilization of the mutant CFTR observed in vivo as compared to in vitro (Supplementary Fig. 3f) may be attributed to the joint effect of entropic stabilization by the crowded cytosolic environment and the coordinated foldase activity of multiple cytosolic chaperone systems[67, 68] including numerous co-chaperones and adapter molecules, providing broader substrate specificity and increased folding capacity than accomplished in the BLM. Comparison of the chaperone-induced functional stabilization of the glucocorticoid receptor (GR)[69] and

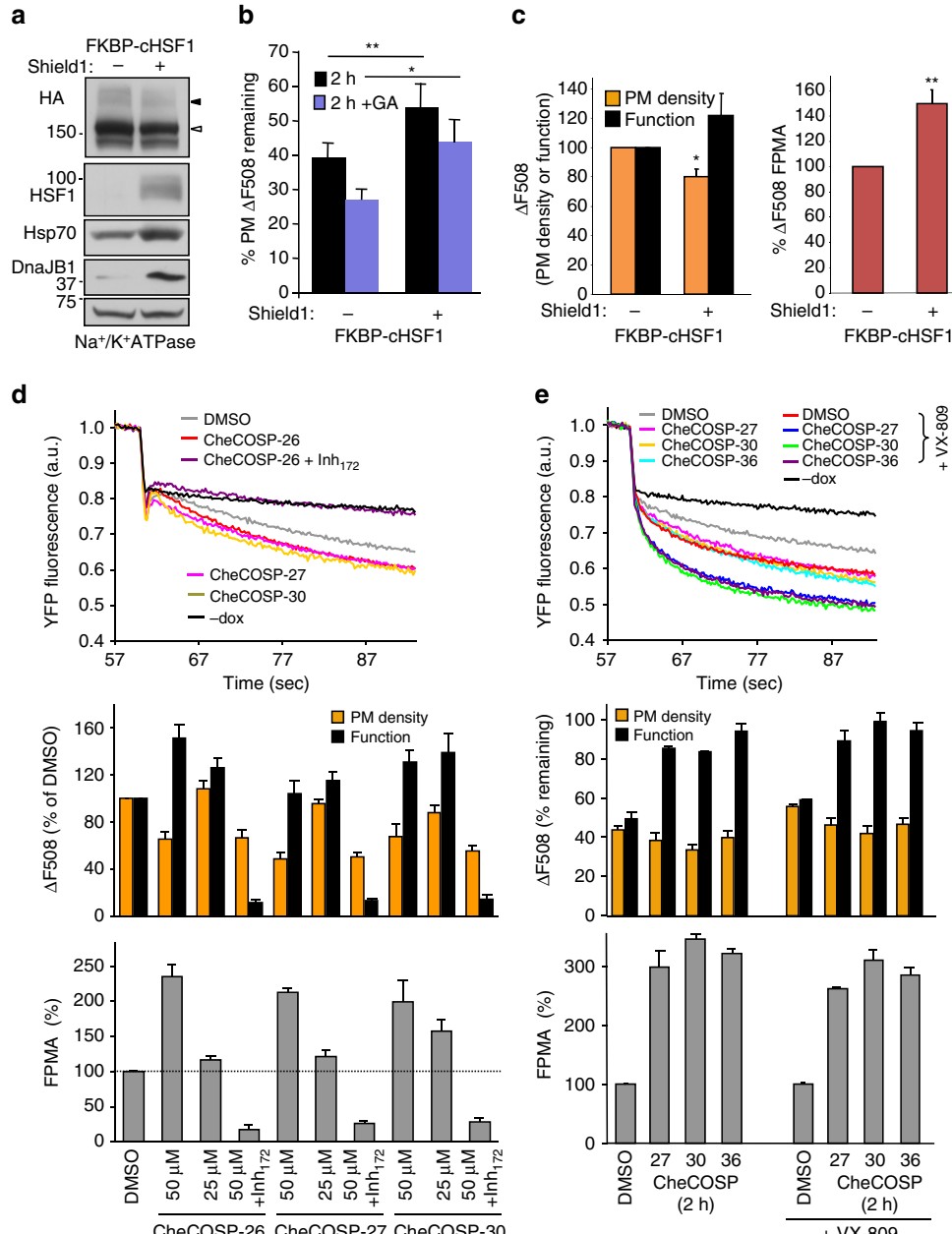

**Fig. 8** Activation of the heat shock response or Hsp90 promotes ΔF508-CFTR function and stability at the PM. **a** Induction of FKBP-cHSF1 and the expression of Hsp70 and DNAJB1 in the presence of 0.2 µM Shield-1 was detected by immunoblotting in HeLa cells, co-expressing ΔF508-CFTR in the presence of VX-809 (3 µM, 37 °C for 24 h). **b** VX-809-rescued ΔF508-CFTR-3HA was stabilized at the PM by cHSF1 induction (0.2 µM Shield-1) in the absence or presence of GA (5 µg ml⁻¹). The CFTR PM density was measured by ELISA after 2 h chase at 37 °C ($n = 3$). **c** The PM density, channel function and FPMA of VX-809 rescued ΔF508-CFTR-3HA were determined in the absence or presence of FKBP-cHSF1 (0.2 µM Shield-1, 37 °C, 24 h) in Hela cells and expressed as percentage of control. CFTR function ($n = 3$) and PM density ($n = 3$) was monitored by iodide efflux assay and ELISA, respectively. **d** The effect of Hsp90 activators on the PM density ($n = 3$) and function ($n = 3$) of ΔF508-CFTR determined by PM ELISA and YFP-quenching assay, respectively. The rescued ΔF508-CFTR (26 °C, 48 h) was unfolded for 2 h at 37 °C in the absence or presence of indicated Hsp90 activators (50 µM CheCOSP 26, 27 or 30). The *lower panel* depicts the FPMA of ΔF508-CFTR. **e** Hsp90 activation stabilizes the functional conformation of the ΔF508-CFTR. The PM turnover ($n = 3$) and functional stability ($n = 3$) of ΔF508-CFTR was determined as in **d**. The rescued ΔF508-CFTR (26 °C, 48 h) was unfolded for 30 min at 37 °C, followed by the activation of Hsp90 (50 µM CheCOSP 27, 30 or 36) during the CHX chase (2 h, 37 °C). VX-809 (3 µM, 24 h) was included as indicated. Data are means ± SEM. *$P < 0.05$ or **$P < 0.01$

temperature rescued ΔF508-CFTR suggests that the Hsc70 chaperone system in addition to its unfoldase activity[70] has substrate specific foldase activity. While the Hsp70/Hdj1 system mediates ligand dissociation and inactivation of the GR via localized unfolding[69], the Hsc70/DNAJ2, similarly to Hsp90/Aha1, was sufficient to partially protect against thermal destabilization of the

mutant in the BLM. In line, inactivation of both Hsc70 and Hsp90 systems was required to unmask the profolding activity of these chaperones on ΔF508-CFTR function and proteases susceptibility in post-Golgi compartments. Furthermore, the tendency to improve the function of thermally unfolded ΔF508-CFTR was improved by the combination of Hsc70/DNAJ2 with

Hsp90/Aha1 as compared to the effect of individual chaperones (Fig. 4d). Additional experimentation is required to assess whether functional coupling of the Hsc70 and Hsp90 folding cycle by HOP and p23 co-chaperones could further enhance the ΔF508-CFTR conformational rescue in the BLM.

Our results expand the previously assigned role of the Hsc70/Hsp90 chaperone systems in the ubiquitin-dependent degradation[13] to the conformational maintenance of non-native PM proteins. At the cellular level the mutant CFTR channel resistance to thermal unfolding, measured by the FPMA, limited proteolysis and PM stability, could be correlated with the folding capacity of the Hsc70/Hsp90, as determined by manipulating the chaperone activity using pharmacological or genetic means. In support, the mutant susceptibility to chaperone-mediated refolding was influenced by the severity of the ΔF508-CFTR conformational defect. Second site suppressor mutations reduced, while destabilization with VX-770 augmented the chaperone susceptibility of ΔF508-CFTR at the PM. The former phenomenon was also reproduced in the BLM. At the single molecule level we provided evidence that molecular chaperones partially buffer the destabilizing consequence of genetic perturbations by remodeling the gating energetics of ΔF508-CFTR, and by extrapolation, of other marginally stable polypeptides (e.g., P67L-CFTR) at the PM[29]. These results, jointly, demonstrate the mutant channel conformational susceptibility to molecular chaperones as an unrecognized determinant of its ion transport activity at the PM.

Based on these results and published data, we postulate that the ultimate fate of marginally stable PM proteins is, at least in part, defined by the joint (re)folding and degradative activity of molecular chaperone networks at the cell surface (Supplementary Fig. 6). Chaperones in coordination with co-chaperones, exchange factors and other conformational sensors can discriminate between the irreversibly and transiently unfolded client protein, a mechanism which complements an array of protein QC steps that evolved for various subcellular compartments[1, 2, 14]. Therefore, the loss-of-function phenotype and disease progression of conformational diseases afflicting PM proteins may be influenced by organ and cell type specific, as well as person-to-person variations in the proteostasis network activity (Supplementary Fig. 6)[1, 2, 71]. Thus, the proteostasis activity is able to tune the phenotypic presentation of conformational diseases by influencing the folding energetic landscape in the context of cell and host physiology, impacting survival and evolvability[1, 2]. These considerations are anticipated to have a potential impact on the utility of proteostasis modulators as therapeutics for human disease.

## Methods

**CFTR expression constructs**. CFTR variants were tagged either with three tandem hemagglutinin epitopes (3HA) in the fourth extracellular loop or a single HA epitope at the C-terminal tail[25, 49]. Nucleotide substitutions to generate CFTR variants (e.g., P67L[33]) and second site mutations were introduced by overlapping PCR mutagenesis as before and primers are listed in Supplementary Table 1[49]. DNAJA1, DNAJA2, Hsc70, and the ATPase deficient Hsc70-K71M variants were expressed by using pProEX-Hta (Clontech) expression vector. The cDNA of FKBP fusion to the constitutively active heat shock factor 1 (FKBP-cHSF1) was a gift of R. L. Wiseman[54].

**Cell lines**. BHK-21 cell (ATTC CCL-10) lines, stably expressing the WT or mutant CFTR variants, were selected in the presence of 500 μM methotrexate[49]. HeLa cells (ATTC CCL2), expressing CFTR-3HA were stably transfected and selected in the presence of 1–5 μg ml$^{-1}$ puromycin as described[34] and transduced with the FKBP-cHSF1 encoding Lenti-X particles to generate double transfected cells. BHK-21 and HeLa cells were grown in DMEM/F-12 (5% fetal bovine serum (FBS, Invitrogen)) and in DMEM (10% of FBS), respectively, at 37 °C in 5% CO$_2$.

The generation of CFBE cell lines expressing inducible CFTR variants has been described previously[34]. Briefly, CFBE41o- (CFBE) is a human CF bronchial epithelial cell line that has a CFTR$^{\Delta F508/\Delta F508}$ genotype[72]. These cells were grown

MEM (Invitrogen) containing 10% FBS (Invitrogen), 2 mM L-glutamine and 10 mM 4-(2-hydroxyethyl)-1-piperazineethanesulfonic acid (HEPES). Plastic flasks for culturing these cells were coated with 10 μg ml$^{-1}$ human fibronectin (EMD), 30 μg ml$^{-1}$ PureCol collagen preparation (Advanced BioMatrix), and 100 μg ml$^{-1}$ bovine serum albumin (Sigma-Aldrich) in LHC basal medium (Invitrogen) (ECM-mix). To generate the inducible expression of CFTR variants, CFBE cells were consecutively transduced with lentiviral particles containing the cDNA for the tetracycline-controlled transactivator and inducible CFTR using the Lenti-X TetON Advanced Inducible Expression System (Clontech). For stable expression the cells were selected with G418 (200 μg ml$^{-1}$; InvivoGen) and puromycin (3 μg ml$^{-1}$, InvivoGen). The Lenti-X Packaging System (Clontech) was used to produce lentiviral particles in HEK293 cells.

**Hsp90 allosteric activators**. The CheCOSP-26 and CheCOSP-27 allosteric Hsp90 ATPase activator synthesis and biological activity on Hsp90 have been previously reported as one of the best Hsp90 ATPase activator, and were designated as compound 18 and 19, respectively in Sattin et al.[55]. CheCOSP-30 and CheCOSP-36 were designed as modifications of CheCOSP27. The chemical synthesis of CheCOSP-30 and CheCOSP-36 has been described[56]. Ongoing experiments show that CheCOSP-30 and CheCOSP-36 activate Hsp90 ATPase and accelerate Hsp90 dimer closure kinetics, as measured by FRET analysis similar to CheCOSP27[55].

**Short-circuit current measurement**. Short-circuit current ($I_{sc}$) measurement of polarized CFBE has been described previously[59]. Briefly, for $I_{sc}$ measurements CFBE cells were grown on ECM-mix coated 12 mm Snapwell filters (Corning) and the CFTR expression was induced for ≥4 days with 500 ng ml$^{-1}$ doxycycline. The Snapwell filters were mounted in Ussing chambers (Physiologic Instruments) in Krebs-bicarbonate Ringer buffer (140 mM Na$^+$, 120 mM Cl$^-$, 5.2 mM K$^+$, 25 mM HCO$_3^-$, 2.4 mM HPO$_4$, 0.4 mM H$_2$PO$_4$, 1.2 mM Ca$^{2+}$, 1.2 mM Mg$^{2+}$, 5 mM glucose, pH 7.4) which was mixed by bubbling with 95% O$_2$ and 5% CO$_2$. Apical NaCl was replaced with Na$^+$ gluconate to generate a chloride gradient and the basolateral membrane was permeabilized with 100 μM amphotericin B (Sigma-Aldrich). After establishing $I_{sc}$ conditions, measurements were performed at 37 °C in the presence of 100 μM amiloride and recorded with the Acquire and Analyze package (Physiologic Instruments). The results are expressed as μA cm$^{-2}$ current density. Measurements were performed in the presence of 100 μM amiloride.

**Reconstitution of CFTR in BLM**. WT- or ΔF508-CFTR variants enriched microsomes were isolated from stably transfected BHK-21 cell lines as described[20, 45]. Briefly, to accumulate temperature-rescued ΔF508-CFTR variants at the PM, transfected BHK-21 cells were cultured at 26 °C for 36 h. To eliminate the ER-associated core-glycosylated channels, cells were exposed to CHX (150 μg ml$^{-1}$) for 14 h at 26 °C before the isolation of microsomes. WT-CFTR expressing cells were cultured at 37 °C and treated with 150 μg ml$^{-1}$ CHX for 3 h at 37 °C. Prephosphorylated CFTR containing microsomes (20–60 μg total protein) were reconstituted from the cis-side of the BLM set-up, containing 0.8–1 ml buffer (300 mM Tris-HCl, 10 mM HEPES (pH 7.2), 5 mM MgCl$_2$, and 1 mM EGTA)[20, 45]. Final ATP concentration in the cup was 2, or 0.5 mM when the effect of 2 mM 2′-deoxyadenosine 5′-triphosphate (dATP) was examined. All experiments were performed in the presence of PKA catalytic subunit (100 U ml$^{-1}$). In the BLM studies, if it is not mentioned otherwise, we used the ΔF508-CFTR-2RK variant.

Current measurements were recorded with a BC-535 amplifier (Warner Instrument, Hamden, CT, USA) and the pClamp 8.1, 9, and 10.3 data acquisition systems (Axon Instruments) at −60 mV holding potential, low pass filtered at 200 Hz by an 8-pole Bessel filter, and digitized at 10 kHz by Digidata 1320 (Axon Instruments). The BLM chamber temperature was raised from 23 to 37 °C at ~1.4 °C min$^{-1}$ rate using the CL-100 temperature controller (Warner Instrument). The success rate of multiple or single ΔF508-CFTR-2RK channel functional reconstitution with complete temperature ramp protocol was ~2 or ~0.7%, respectively, in the BLM. The functional incorporation success of ΔF508-CFTR and ΔF508-CFTR-2RK into the BLM was ~1 and ~2 %, respectively, based on a ~300 ΔF508-CFTR and ~1800 ΔF508-CFTR-2RK incorporation attempts.

**Energetic analysis of the CFTR gating cycle**. Records were digitally filtered at 50 Hz by using Clampfit 10.3 (Axon Instruments) and events were idealized using a half amplitude-threshold. Events shorter than 10 ms were rejected from further analysis. The single channel open probability ($P_O$) was determined using the event detection features of the Clampfit 10.3 software, as $NP_O$ divided by the number of channels ($N$).

Single channel dwell times were determined from histograms. We used Gaussian distribution with nonlinear least-squares Levenberg–Marquardt algorithm for dwell time histogram fitting. Histograms were created from open and closed dwell times using logarithmic x-axes with 10 bins per decade, using the Clampfit10.3 software. Analysis of the gating kinetics and thermodynamics was performed from single channel current records. Activation enthalpy and entropy of single channel gating transitions were calculated from opening and closing rate constants [$k_O = (\tau_C^{-1})$ and $k_C = (\tau_O^{-1})$, respectively] according to the equation[73]

$$\ln(k) = -\Delta H^{\ddagger} R^{-1} T^{-1} + \Delta S^{\ddagger} R^{-1} + \ln \nu^{\ddagger},$$

where $R$ is the gas constant (8.31451 J mol$^{-1}$ K$^{-1}$), $\Delta H^{\ddagger}$ is the activation enthalpy, and $\Delta S^{\ddagger}$ is the activation entropy for either channel opening or closing. We used $\nu^{\ddagger} \sim 10^{6}$ s$^{-1}$ as described for protein folding[73]. One or two component fits were calculated by linear regression analysis for $\ln(k_O)$ and $\ln(k_C)$ plotted against $T^{-1}$. $\Delta H^{\ddagger}$, $\Delta S^{\ddagger}$, and $\Delta G^{\ddagger}$ were calculated from the Arrhenius equation ($\Delta G^{\ddagger}_{C-O} = \Delta H^{\ddagger}_{C-O} - T\Delta S^{\ddagger}_{C-O}$ and $\Delta G^{\ddagger}_{O-C} = \Delta H^{\ddagger}_{O-C} - T\Delta S^{\ddagger}_{O-C}$, see Fig. 6b and Supplementary Table 4). The Gibbs free energy of single channel open states was calculated from the activation energies of opening and closing according to $\Delta G_{O-C} = \Delta G^{\ddagger}_{O-C} - \Delta G^{\ddagger}_{C-O}$.

To estimate steady-state gating energetics using both single and multichannel recordings, the $\ln K_e$ values (where $K_e$ is the equilibrium gating constant and $K_e = P_o (1 - P_o)^{-1}$) were plotted as a function of $1000T^{-1}$ (K$^{-1}$). The $\Delta H_{O-C}$ and $\Delta S_{O-C}$ were calculated from Van't Hoff plots;

$$\ln K_e = -\Delta H_{O-C} R^{-1} T^{-1} + \Delta S_{O-C} R^{-1}.$$

The Gibbs free energy of the open state relative to the closed state was derived from the equations of $\Delta G_{O-C} = -RT \ln(K_e)$, where $R$ is the universal gas constant and $T$ is the absolute temperature, and $\Delta G_{O-C} = \Delta H_{O-C} - T\Delta S_{O-C}$.

**CFTR limited proteolysis.** Limited proteolysis was performed published previously[50]. Briefly, WT or ΔF508 CFTR containing microsomes were digested with protease in PBS for 15 min at 4 °C. To terminate the proteolysis, 1 mM phenylmethylsulfonylfluoride (PMSF) and 5 mM EDTA was added followed by denaturation with 2× Laemmli sample buffer at 37 °C for 10 min.

**Immunoblotting and immunoprecipitation.** Immunoblotting of cell lysates and immunoprecipitates (IP) were performed as previously described[24]. Anti-ErbB-2 (9G6, #16899) was purchased from Abcam (Cambridge, MA). Co-IP of chaperons with WT-, ΔF508-, and rescued ΔF508-CFTR-HA (24 h at 26 °C and 5% glycerol) was performed after solubilization of BHK-21 cells in 150 mM NaCl, 20 mM Tris 0.1% NP-40, pH7.4 and protease inhibitors 5 μg ml$^{-1}$ leupeptin, 5 μg ml$^{-1}$ pepstatin, 1 mM PMSF and 10 mM iodoacetamide. A total of ~5 mg BHK-21 lysate was used for each Co-IP. The lysate was incubated with anti-HA antibody for 1 h and then CFTR-antibody complexes were adsorbed on 80 μl of 50% protein G-beads (GE Healthcare, Uppsala, Sweden) for 1 h at 4 °C. The beads were washed three times and proteins were eluted in 2× Laemmli sample buffer (20 min, 37 °C). Co-IP of co-chaperones with CFTR was performed after cross-linking with 0.1 mM dithiobis[succinimidyl propionate] (DSP), of cells exposed to CHX chase at 26 or 37 °C to eliminate the core-glycosylated forms as described for Fig. 1a. Prior to SDS-PAGE, DSP was reduced with 100 mM DTT. The antibodies were used: HA (HA.11, clone 16B12, Cederlane), Hsc70 (SPA-815, Stressgen), Hsp40 (SPA-400, Stressgen), and Hsp90 (CA1016, Calbiochem) (Supplementary Table 5).

**CFTR PM density measurement.** CFBE cells were seeded on extracellular matrix-mix coated 12 mm Snapwell filters (Corning) at a density of $1 \times 10^5$ cells cm$^{-2}$. CFTR expression was induced with 250–500 ng ml$^{-1}$ doxycycline (dox) for 4–5 days. The PM density and stability of CFTR-3HA in BHK-21 and CFBE cells was measured by cell surface ELISA[24], using anti-HA antibody (MMS101R, Covance) and secondary HRP-conjugated goat anti-mouse IgG antibody (GE Healthcare) in PBSCM (phosphate buffered saline with 0.1 mM CaCl$_2$ and 1 mM MgCl$_2$ at pH 7.4) supplemented with 0.5% bovine serum albumin on ice as described[24]. Excess antibody was removed by extensive washing and specific binding was determined with the Amplex-Red (Invitrogen) as HRP substrate. The fluorescence intensity was measured at 544 nm excitation and 590 nm emission wavelengths with a POLARstar OPTIMA (BMG Labtech) fluorescence plate reader[24].

**Iodide efflux assay.** CFTR mediated iodide efflux was determined in transfected HeLa cells[74]. Briefly, CFTR was activated by PKA agonist-cocktail containing 20 μM forskolin, 0.25 mM 8-(4-chlorophenylthio)-adenosine 3′,5′-cyclic monophosphate (cpt-cAMP), 0.5 mM 3-isobutyl-1-methylxanthine (IBMX) and 50 μM genistein. The iodide efflux into the extracellular compartment was monitored with an iodide-selective electrode. CFTR-mediated iodide transport was calculated from peak value of iodide release after normalizing for protein content. The efflux rate of NT siRNA transfected cells was used as negative control[24].

**Halide-sensitive yellow fluorescent protein assay.** CFTR function measurement by the halide-sensitive yellow fluorescent protein assay has been described previously[59]. Briefly, CFBE cells harboring the inducible expression of ΔF508-CFTR were generated to co-express the halide sensor YFP-F46L/H148Q/I152L by lentiviral transduction. Double-expressing cells were enriched by fluorescence-activated cell sorting. YFP-expressing cells, seeded onto 96-well microplates at a density of $2 \times 10^4$ cells per well, were induced for ΔF508-CFTR expression for 2–4 days at 37 °C, in some cases followed by low temperature—rescued for an additional 48 h at 26 °C. During the assay the CFBE cells were incubated in 50 μl per well phosphate-buffered saline (PBS)–chloride (140 mM NaCl, 2.7 mM KCl, 8.1 mM Na$_2$HPO$_4$, 1.5 mM KH$_2$PO$_4$, 1.1 mM MgCl$_2$, 0.7 mM CaCl$_2$, and 5 mM glucose, pH 7.4). CFTR was activated by 50 μl well-wise injection of activator solution (20 mM forskolin, 0.5 mM IBMX, 0.5 mM cpt-cAMP, 100 μM genistein). Then the quenching reaction was started by the injection of 100 μl of PBS-iodide, in which NaCl was replaced with NaI, while the YFP-fluorescence was recorded at 485-nm excitation and 520-nm emission for 36 s at a 5-Hz acquisition rate in a POLARstar OPTIMA (BMG Labtech) fluorescence plate reader. Background values were subtracted and the YFP signal was normalized the fluorescence before NaI injection. The iodide influx rate was calculated by linear fitting to the initial slope.

**Conformational stability measurements.** Differential scanning fluorimetry of isolated NBD1 variants (7–12 mM) was performed in 150 mM NaCl, 20 mM MgCl$_2$, 10 mM HEPES, and 2.5 mM ATP at pH 7.5 in the presence of 4× Sypro Orange, essentially as described[20, 75, 76]. Unfolding was monitored between 25 and 70 °C with the Stratagene Mx3005p (Agilent Technologies, La Jolla, CA, USA) and analyzed by using XLFit (IDBS, London, UK). Alternatively, melting curves were recorded between 10 and 70 °C with the QuantStudio7 Flex qPCR machine (Life Technologies, Carlsbad, CA, USA) and analyzed using the Protein Thermal Shift Software v1.0 (Applied Biosystems, NY, USA). In both cases the temperature ramp rate was 0.017 °C s$^{-1}$ and data were fitted with a Boltzmann sigmoidal function.

Following chaperone inhibition, the conformational stability of ΔF508-CFTR in its native environment was probed by limited tryspinolysis and immunoblotting[50], using the 660 NBD1 specific monoclonal antibody, kindly provided by Dr. J. Riordan and the CFFT[20]. Microsomes were isolated from stably transfected BHK-21 cells after low temperature rescue (26 °C, 48 h) and unfolding (37 °C, 1.5 h) in the presence of DMSO or pifithrin-μ (Pif, 5 μM) and GA (5 μg ml$^{-1}$)[50]. To eliminate endogenous core-glycosylated CFTR, BHK-21 cells were treated with CHX (150 μg ml$^{-1}$) during the 37 °C incubation. Limited proteolysis was performed at the indicated trypsin concentrations (Worthington Biochemical, Lakewood, NJ, USA) for 15 min on ice. Trypsinolysis was terminated by 10 mg ml$^{-1}$ soybean trypsin inhibitor (Sigma-Aldrich), 10 mM PMSF and 2× Laemmli sample buffer. The remaining amount of the trypsin-resistant complex-glycosylated ΔF508-CFTR was quantified by densitometric analysis of anti-HA immunoblots with multiple exposures using a DuoScan transparency scanner and the NIH ImageJ v1.6 software (NIH, http://rsb.info.nih.gov/ij/) and expressed as the percentage of the initial amount.

**Luciferase refolding assay.** DnaK, DnaJ, and GrpE were expressed and purified by conventional techniques using expression plasmids (pDS56-dnaK-Chis6, pUHE21-2fdD12-dnaJ, pZE2-Pzl-grpE) and strains (BB1553, W3110), kindly provided M Mayer according to published protocols[77–79]. Recombinant Firefly luciferase (Sigma-Aldrich) refolding experiments were performed as described[42]. Briefly, following the denaturation of Firefly luciferase (8 μM) in 6 M Gu-HCl (5 min, 26 °C), refolding was initiated by 100-fold dilution into 40 mM HEPES/KOH pH 7.5, 50 mM K-acetate, 2 mM Mg-acetate, and 2 mM ATP at 30 °C in the absence or presence of the indicated chaperone and co-chaperone DnaK (800 nM), DnaJ (160 nM) and GroE (400 nM). Luciferase activity was measured as a function of incubation time by luminometry in the presence of 70 μM D-luciferin and 5 mM ATP.

**Vesicular pH measurements by FRIA.** The postendocytic fate of low-temperature rescued ΔF508-CFTR after internalization from the PM was monitored by fluorescence ratio image analysis (FRIA) in CFBE cells[80]. The FRIA of endocytic vesicles containing CFTR cells was preformed as before[80]. Briefly, filter-grown CFBE was allowed to polarize for 5 days and temperature rescued for 48 h at 30 °C. DMSO or VX-809 was added for 24 h and kept during the experiment. GA (5 μM) and Pif (5 μM) were added to the apical site of Transwell chamber during anti-HA antibody incubations on ice, and kept at 37 °C during the chase. Then, cells were labeled by FITC-conjugated goat anti-mouse secondary Fab (Jackson ImmunoResearch) on ice. Internalization was allowed for 30 min at 37 °C. FRIA was determined by an AxioObserver Z1 (Carl Zeiss MicroImaging) inverted fluorescence microscope equipped with an X-Cite 120Q fluorescence illumination system (Lumen Dynamics Group) and an Evolve 512 EMCCD (electron-multiplying charge-coupled device) camera (Photometrics Technology) at 495 ± 5 nm and 440 ± 10 nm excitation and at 535 ± 25 nm emission wavelength. MetaFluor (Molecular Devices) software was used for data acquisition and analysis.

**siRNA transfection.** Hsc70 was knocked down (KD) in HeLa cells, stably expressing WT- or ΔF508-CFTR-3HA, by using 50 nM GENOME SMARTpool siRNA (siHsc70) (Dharmacon-Thermo Fisher Scientific, Rockford, IL) and Oligofectamine (Invitrogen) according to the manufacture's instructions. Non-targeting (NT) siRNA (D-001210-01, Dharmacon) was used as a negative control. KD efficiency was verified by quantitative immunoblot analysis after 4 days of siRNA transfection. Low temperature rescued ΔF508-CFTR (26 °C for 36 h) was unfolded for 2.5 h at 37 °C in the absence or presence of chaperone inhibitors as indicated.

siGENOME SMARTpool or single siRNA (Dharmacon-Thermo Fisher Scientific, Rockford, IL) listed in Supplementary Table 5 were transiently transfected into CFBE cells using RNAiMAX reagent (Thermo Fisher Scientific) according to the manufacturer's instructions. Cells were trypsinized and seeded into 24 well plates 5 h before siRNA transfection. Knockdown efficiency of the target proteins was verified by quantitative immunoblot analysis. Experiments were

performed 4 days after siRNA transfection. Detailed information on siRNAs and antibodies are described in Supplementary Table 5. In most case, the KD efficiency was > 60%.

**Immunostaining**. Immunostaining of CFTR and HSF-1 was performed in ΔF508-CFTR-3HA and FKBP-cHSF1 expressing HeLa cells. HeLa cells were fixed with 4% paraformaldehyde and permeabilized with 0.2% Triton X-100, followed by blocking with 0.5% BSA in PBS, supplemented with 0.1 mM $CaCl_2$ and 1 mM $MgCl_2$, pH 7.4 (PBSCM). FKBP-cHSF1- and ΔF508-CFTR-3HA-expressing HeLa cells were costained with anti-HA (dilution 1:1000) and anti-HSF1 (NB300-730, NovusBio, dilution 1:1000) antibodies. We used Alexa Fluor-488 and Alexa Fluor-555 conjugated secondary antibodies (Invitrogen). Images were obtained on an LSM780 laser confocal fluorescence microscope (Carl Zeiss, Jena, Germany).

**Statistical analysis**. Results are presented as means ± SEM for the indicated number of experiments as biological replicates. Statistical analysis was performed by two-tailed, unpaired Student's *t*-test with the means of at least three independent experiments and ≥95% confidence level was considered significant. The normal distribution was validated by calculating the skew factor and analyzing the data on a normal probability plot. If this condition could not be confirmed for some of the BLM data, Mann–Whitney *U* test was used for calculating the *P*-values (www.socscistatistics.com/tests/mannwhitney/Default2.aspx).

To ensure acceptable reproducibility, biological replicates were performed and the number of independent experiments is indicated in each figure legend or in the figure. Depending on the specific assay, the number of technical replicates in individual experiments was as follows. Short circuit current measurement, two; halide-sensitive YFP transport measurement, four; iodide-selective chloride transport measurement, three; PM CFTR density determination by ELISA, two to four; luciferase refolding, two or three, and melting assay, three. In each FRIA studies several hundreds of vesicles were analyzed from several cells in the indicated number of independent experiments. For limited proteolysis, immunoblots/IP and bilayer measurements technical replicates for feasibility reasons were not used and the number of biological replicates is indicated in the figure legends.

**Data availability**. All data supporting the findings of this study are available from the corresponding author upon reasonable request.

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

## Acknowledgements

We thank J. Riordan, T. Jensen and the Cystic Fibrosis Foundation Therapeutics for generously providing the 660 Ab, M. Sharma for conducting pilot studies, D. Gruenert for providing the CFBE14o- cells, R.L. Wiseman and A.S. Verkman for the gift of FKBP-cHSF1 and YFP-F46L/H148Q/I152L constructs, respectively, M. Mayer for providing the prokaryotic DnaK/DnaJ/GrpE expression system and valuable advices, L. Bene and A. Aleksandrov for valuable advices. G. Veit, I. Baaklini and R. Fukuda were supported in part by FRSQ, GRASP and JSPS postdoctoral fellowships, respectively. G. Colombo received funding from Fondazione Cariplo #2011.1800, Premio fondazione cariplo per la ricerca di frontiera, from Associazione Italiana Ricerca sul Cancro, IG 15420; and Universita' degli Studi di Milano is acknowledged for a grant to S.S. (Assegno di ricerca tipo A). This work was supported by grants from the CIHR, CF Canada, NIH-NIDDK, CF Foundation Therapeutics Inc., and the Canadian Foundation for Innovation for W.B., J.C.Y. and G.L.L. J.C.Y. and G.L.L. are holders of Canada Research Chairs.

## Author contributions

M.B., G.V., R.F., R.G.A., T.O., G.S., H.X., P.M.A., L.K.B. and A.R. performed the research and analyzed the data, I.B. and J.S. purified the chaperones, S.S. and G.C. synthesized the Hsp90 modulators, W.B. and J.C.Y. edited and revised the manuscript, G.L.L. designed the research and wrote the article.

## Additional information

**Competing interests:** The authors declare no competing financial interests

