## [Peer Review file · Nature Communications]

Reviewers' comments:

Reviewer #1 (Remarks to the Author):

This manuscript explores a role for the Hsp70, Hsp90 and their co-chaperones in the regulation of CFTR un-folding. The authors take advantage of the fact that folding/function of CFTR deltaF508 is approximately normal at 26C, but 2 hours at 37C reveals folding defects and dysfunction. Another key design of the experiments, especially early in the paper, is to measure CFTR levels and activity independently, along with separating CFTR inhibitor sensitive and PKA-stimulated activity – to help account for effects on CFTR turnover, other channels and function. They also compare deltaF508 CFTR to WT to focus on roles played in the unstable, CF-associated channel. In HeLa cells, they find that Hsc70 knockdown increases CFTR deltaF508 density (presumably by blocking turnover), while causing a mild reduction in active channel (presumably because the material left is not properly folded). One of the strengths of the paper is the next set of studies, in which the authors reconstitute CFTR into vesicles (BLMs) and test the effects of added chaperones. The results suggest that Hsc70-DnaJA2 and Hsp90-Aha1 are able to partially rescue deltaF508 function, perhaps by stabilizing (or permitting) the open state. With a couple of assumptions about the number of states, they proceed to calculate free energy landscapes in this system to reveal the effects of chaperones on kinetic and thermodynamic stability. The assumptions that lead to the free energy calculations are probably optimistic and the model does not seem to be readily addressable by a distinct method, such as FRET, but the exercise is still likely to be clarifying because it illustrates an interesting hypothesis that matches intuition and is consistent with the experimental observations. The authors set out to explore a very challenging topic and have used state-of-the-art methods to do so. The results are provocative and point to a role for the stress response in cystic fibrosis.

A few concerns:

1. After reviewing the data a few times, the role of Hsp90 in deltaF508 maintenance is still rather mysterious. The Hsp90 inhibitor seems to have little effect by itself and Hsp90 protein (even with Aha1) has a modest effect in the BLM studies (compared to Hsc70-DnaJA2). To this reviewer, the totality of the data suggests that Hsp90 might play a secondary (or even no) role in CFTR maintenance. Does siRNA of Hsp90alpha or Hsp90beta impact PM density and FPMA in the HeLa cells? Does Aha1 knockdown impact FPMA?
2. There are a couple of major paralogs of the Hsp70 family that seem likely to interact with CFTR: the two major cytoplasmic forms: Hsc70 (HSPA8) and Hsp70 (HSPA1A) and the ER-resident form BiP (HSPA5). Sometimes, paralogs have opposing effects on protein stability and folding (Jinwal et al. 2013 FASEB J), and Hsp70 (HSPA1A) is more often associated with pro-degradation functions. In that context, it seems important to understand whether siRNA of HspA1A (and possibly HSPA5) have the same phenotype as HSPA9. The inhibitors are likely not paralog selective, so differences could account for the larger effect of inhibitors on FPMA than siRNA of HSPA8. Also, it seems important to check whether the siRNA is selective for Hsc70, as the paralogs are very closely related. It is also useful to check if Hsc70 or Hsp90 knockdown is affecting the levels of DnajA2, Aha1, Hsp70 (HSPA1A) and Bip (HSPA5).
3. This work represents a tour-de-force in experimental design and implementation, yet some key links between the experimental paradigms (HeLa cells, BLMs) remain unresolved – sometimes giving a patchwork feeling to the inquiries. As mentioned above, what is the effect of Aha1 knockdown and simultaneous Hsp90/Aha1 knockdown in the HeLa cells (Fig 1)? Effects of DnaJ1, DnaJA2 knockdown? Effects of Hsc70 and Hsp90 inhibitors (and combinations) in the BLMs (Fig 2)? Many of the chaperone effects, especially individually, are somewhat subtle (statistically significant, but with activity well below WT), so it seems imperative to make sure that the effects are consistent across the different experimental paradigms.
4. Hsc70 binds tightly to lipids and is known to stabilize lysosomes (Arispe 2002 Cell Stress Chaperones 7:330). It isn't clear how much of the effect of Hsc70 on deltaF508 CFTR might arise from direct interactions with the channel and how much is due to important, but indirect, effects

on lipid dynamics. Either outcome is interesting, but it seems that the current work would benefit from some clarity on this issue. One way to test it might be to add peptide, NRLLLTG, a well-known competitive ligand of client binding (Zhang et al. 2014 PloS One 9:e103518) that doesn't interfere with lipid binding.

Minor:

5. DnaJA1 and DnaJA2 are known to have different effects on luciferase refolding (Tzankov 2008 JBC 283:27100, Rauch et al 2014 JBC 289:1402), with DnaJA1 being inactive. This is consistent with the author's observations on the failure of DnaJA1 to restore CFTR in the BLM studies.

6. The experiments with DnaK, DnaJ and GrpE (pg 12-13) are nice, but it isn't clear if they add anything to the narrative.

Reviewer #2 (Remarks to the Author):

The authors study the impact of chaperones on the folding energetic landscape of $\Delta F508$ CFTR. They investigated the effect of chaperone activity on the functional and conformational maintenance of $\Delta F508$ -CFTR at the PM and following its reconstitution in phospholipid bilayer. They found that inactivation of the temperature-rescued, native-like $\Delta F508$ -CFTR upon thermal unfolding was partially suppressed by Hsc70- and Hsp90-cochaperone both in vivo and in vitro. This was accomplished by kinetic and thermodynamic remodeling of the gating energy landscape of single molecules toward the wild-type channel, an effect mimicked by second-site suppressor mutations. They conclude that chaperones contribute to functional adaptability of $\Delta F508$ -CFTR by reshaping the final fold conformational energetic, a mechanism with implications in influencing (re)folding of ABC-transporters and other PM proteins in health and diseases.

Major criticisms

1. Although the authors should be complemented on a large body of work, the manuscript is so densely written and filled with jargon that it will be difficult to have an impact on a general audience.

2. The authors argue that they can address a question of folding by evaluating the effect of unfolding, ie thermal stability. Yet there is no experimental evidence to support this claim.

3. The authors begin the results with supplemental data. The authors should put all relevant data in the body of the manuscript.

4. The authors measure FPMA a complex measurement of channel activity vs. PM density monitored with PM Elisa. How was this term validated? For example ablation of Hsc70 increase density but less than channel activity leading to a decrease in FPMA. Was Hsc70 ablated as mentioned? The effects of the drugs is very confusing and explained by overlapping roles of 70 and 90. Finally Fig 1C shows nearly identical results between the DMSO and the drug treated groups. A small effect was only noted at high doses.

5. Why was the mutant P67L chosen? Seems out of place and the drugs only had a small effect.

6. The authors demonstrate conformational resistance with VX-809 in Fig 1H. Do the authors know that this Hsc70 inhibitor is still working in the presence of VX-809? It is possible that VX-809 is having a direct effect on Hsc70/Hsp90 and thus the Pif+GA combination would have less of an effect.

7. The results on thermal inactivation of the open probability are highly confusing. For example the revertant mutants never reach wt level 2F, even with the addition of Hsc70 or 90 plus the co-chaperones. Is there any evidence that protection from thermal inactivation is directly related to conformational remodeling? What if the chaperones bind to the NBD domains and increase open probability directly?

8. In Fig 3A why is wt-CFTR so different from $\Delta F508$ in all the experiments. One would predict that they would be closer.

Reviewer #3 (Remarks to the Author):

The manuscript of Bagdany et al. contains an impressive number of experiments with a wide range of biochemical techniques and functional characterizations. They demonstrate in detail that the folding defect (and consequently, the functional defect) of $\Delta F508$ -CFTR can be partially rescued by lowered temperature, chaperones and other stabilizing agents.

The core of this work is the detailed characterization of the gating energy landscape and how it is influenced by unfolding of the mutants during temperature ramps and the different methods of rescue. I find this approach very interesting. However, a number of issues with the presentation and analysis of the data left me thoroughly confused. I cannot decide from the manuscript whether the data are sufficient to support the conclusions. I can therefore not recommend the publication at this stage. I have only listed the more serious issues. My comments below are grouped by topic rather than importance.

Biochemical experiments

Page 9: "These suppressor mutants conferred wt-like resistance to $\Delta F508$ -CFTR upon inhibition (by Pif+GA) as indicated by the large unaltered FPMA values (Fig. 1F)." Figure 1F does not prove that the inhibitors have no effect since the control is missing.

cHSF1 increased PM expression in Fig. 7B and decreases it in Fig. 7C. This apparent contradiction is never mentioned in the text.

Thermodynamic analysis

What is the estimated systematic error introduced by the assumption of a system with only two states (O-C) for when the system actually has more states (at least two closed and two open, according to the dwell time histograms)?

Page 12/figure 3B/C: The change in ΔG when adding chaperones to $\Delta F508$ -2RK in planar lipid bilayers is attributed solely to the change in $T\Delta S$. However, ΔH undergoes about the same dramatic change, both parameters switch from positive to negative, but none of this is mentioned in the text. Does the channel switch between enthalpic and entropic stabilization under different conditions?

Dwell time histograms

The legends of Fig. 4B/S4A state that you fitted the dwell time histograms with Gaussian distributions instead of exponentials as it is done by virtually every other publication on single-channel recording. If that is true and not a typing error, this decision needs to be explained.

However, are Fig. 4B/S4A actually standard log-binned dwell time histograms? I am not sure. The text on page 13 3rd sentence after the paragraph heading speaks of dwell time histograms. The figure legends speak of "cumulative open/closed time histograms" instead. Dwell time histograms and cumulative dwell time histograms are two different things. And since cumulative dwell time histograms are to my knowledge not often used in single-channel analysis, they should be explained a bit.

Do Fig. S4A and 4B display the same type of histogram? One has log(open time) on the x-axis, the other log(mean closed time). The latter is highly unusual, please explain.

In summary, I cannot figure out what is displayed here, since the histograms are not explained in the methods.

Discussion

On the bottom of page 20, the authors mention the "folding landscape of gating". Do they suggest that opening and closing of the channel are large-scale folding events? This would be quite provocative and needs a thorough explanation, instead of just being implicated in passing.

Presentation

The way the article is written made it difficult for me to read. The main problem I had were not the numerous grammar errors, those could easily be fixed. But some sentences were worded in such a way that I had genuine difficulties understanding them, I suspect that some are even wrong. Other

things are downright sloppy, like wrong figure descriptions and a paragraph heading that randomly repeats in the middle of the text. In summary, the manuscript left me with the impression that it was not proofread by the authors.

I'm listing some of the more confusing examples below:

- Introduction, page 3, 1st paragraph : "... bind to partially folded intermediates and account for maintaining their active conformation." Do the authors imply that those intermediates already active or should that read "reach" rather than "maintain"?
- Introduction, page 3, 1st paragraph: That the buffering capacity of chaperones with regards of folding proteins with random mutations enhances genetic diversity is plausible. But how does buffering for environmental insults and errors in protein synthesis increase the genetic diversity?
- The sentence between pages 3 and 4 contains no less than 70 words.
- Results, page 5, the long sentence at the end of the 2nd paragraph. The digestion patterns of partially folded wt and unfolded mutant are similar. The authors conclude from this that it is plausible that the intermediates of the mutant during folding and refolding are similar to each other. I don't understand this conclusion.
- Legend to Figure S1A: "...was either unfolded for 2.5h at 37° (the main text referring to this figure on page 6 says 2 hours) in the presence of ... (lanes 3-6) or exposed to 37° for 20 min and then cultured at 26° for 12 hours". Where is this second condition shown and why was it done?
- Legend to Figure S2B: "Association of Hsc70, but not Hsp90 with CFTR was dependent on the ATP concentration." I don't understand this statement. The dark grey bars with ATP do not differ substantially from the red bars without ATP. The association seems to depend on "Apy" instead.
- Figure 1A has "FPMA" on the y-axis, but displays, according to the legend the PM density and the I- efflux individually, not their ratio.
- The legend to Figure 1A,B states 2.5 hours of heat stress, the text referring to the figure states 2 hours.
- The empty arrowhead in Fig. 1C is only explained somewhere in the Supplements.
- Page 8, referring to Fig. 1F: "Inhibition of Hsc70/Hsp70 and Hsp90 in combination with Pif+GA ...". There is no hint in the figure of what additional chaperone inhibition was combined with Pif and GA. Did the authors perhaps mean "Inhibition by a combination of Pif+GA"?
- Page 10, first sentence: "We reconstituted the mutant inactivation process in BLM." How does one reconstitute a process? Should the sentence read "We reconstituted the mutant into BLMs to observe the inactivation process?"
- The legend to Figure S2D states "WT-CFTR activity was monitored at ~34° for ~6 min". The figure itself is labelled with a temperature gradient from 27.8° to 35.2° and back to 34.4°. To my estimate, the time spent between 34° and 35° could be what the authors meant in the legend. Please clarify.
- Page 12, last paragraph. What is "unfolding gating kinetics"? Do you mean unfolding kinetics or gating kinetics?
- Materials and Methods: the paragraph heading "Reconstruction of CFTR..." is repeated in the middle of the text for no apparent reason.

Reviewer #1

...The assumptions that lead to the free energy calculations are probably optimistic and the model does not seem to be readily addressable by a distinct method, such as FRET, but the exercise is still likely to be clarifying because it illustrates an interesting hypothesis that matches intuition and is consistent with the experimental observations. The authors set out to explore a very challenging topic and have used state-of-the-art methods to do so. The results are provocative and point to a role for the stress response in cystic fibrosis.

We are glad that Reviewer#1 recognized the importance and challenges of our work and we much appreciate the valuable suggestions provided.

A few concerns:

1. After reviewing the data a few times, the role of Hsp90 in deltaF508 maintenance is still rather mysterious. The Hsp90 inhibitor seems to have little effect by itself and Hsp90 protein (even with Aha1) has a modest effect in the BLM studies (compared to Hsc70-DnaJA2). To this reviewer, the totality of the data suggests that Hsp90 might play a secondary (or even no) role in CFTR maintenance. Does siRNA of Hsp90alpha or Hsp90beta impact PM density and FPMA in the HeLa cells? Does Aha1 knockdown impact FPMA?

We agree that effect of Hsp90 was somewhat less prominent than that of Hsc70 in the BLM studies. Nevertheless, Hsp90/Aha1 activity has increased the P_o by ~1.5-2-fold of the mutant channel at 30-36°C. We also agree that specific inhibition of Hsp90 by geldanamycin (GA) has no effect on the Fractional Plasma Membrane Activity (FPMA) of mutant CFTR in HeLa cells (original Fig.1B). This could be explained, in part, by the over-expression level of Hsp70 chaperone family members (at least six of them confined to the cytosol, e.g. Hsp70 and Hsp70-2), a hallmark of tumors and tumor derived cell lines (*Genes Dev* 2005, 19: 570-582), represented by HeLa cells. The increased concentration of Hsp70 family members in the cytosol may compensate for the GA-induced loss of Hsp90 α and Hsp90 β refolding activity in HeLa cells. In line, upon ablation of Hsc70 by siRNA, GA significantly reduced the mutant channel Fraction Plasma Membrane Activity (FPM) from $90.9 \pm 1.9 \%$ to $74.5 \pm 2.3 \%$ ($p < 0.01$). Similar results were obtained by using pharmacological inhibition of Hsc70/Hsp70 (Apoptozole or Pifithrin) in combination with GA (original Fig.1B and revised Fig.S1G).

Evidence supporting more a prominent role of the Hsp90 was obtained in CFBE, a highly relevant cellular model of human respiratory epithelia. Acute inhibition of Hsp90 by GA alone reduced the FPMA of $\Delta F508$ -CFTR significantly more than observed for the WT ($67 \pm 3.6\%$ versus $82 \pm 4.8\%$, $p < 0.05$, respectively, revised Fig.2D). Furthermore, as requested by the Reviewer, we tested the ablation effect of Hsp90 α , Hsp90 β and Aha1 on the mutant FPMA in CFBE cells. Small, but significant reduction of the $\Delta F508$ -CFTR FPMA was observed for Hsp90 α , Hsp90 β and Aha1, suggesting the contribution of the Hsp90 chaperone system activity to the $\Delta F508$ conformational maintenance. These new results were included as Fig. 2B.

Collectively, these observations suggest that the stand-alone effect of Hsp90 inhibition in CFBE could be attributed to the differential activity of proteostasis machinery in HeLa and CFBE cell lines. The CFBE was generated by transformation of *native* cystic fibrosis (CF) tracheo-bronchial cells with SV40 (*Cell Tissue Res.* 2006 323 :405-15), containing lower concentration of molecular chaperones than the tumor derived HeLa cells, with significantly elevated Hsp/Hsc70 chaperoning activity (*Genes Dev.* 2005, 19: 570-582).

In light of these results and considerations, we decided to de-emphasize the physiologically less relevant HeLa results and move them into the Supplement, while adding the new siRNA chaperone/co-chaperone screening results on CFBE cells as Fig.2B and Fig. S1A-C.

2. There are a couple of major paralogs of the Hsp70 family that seem likely to interact with CFTR: the two major cytoplasmic forms: Hsc70 (HSPA8) and Hsp70 (HSPA1A) and the ER-resident form BiP (HSPA5). Sometimes, paralogs have opposing effects on protein stability and folding (Jinwal et al. 2013 FASEB J), and Hsp70 (HSPA1A) is more often associated with pro-degradation functions. In that context, it seems important to understand whether siRNA of Hspa1A (and possibly HSPA5) have the same phenotype as HSPA9. The inhibitors are likely not paralog selective, so differences could account for the larger effect of inhibitors on FPMA than siRNA of HSPA8. Also, it seems important to check whether the siRNA is selective for Hsc70, as the paralogs are very closely related. It is also useful to check if Hsc70 or Hsp90 knockdown is affecting the levels of DnajA2, Aha1, Hsp70 (HSPA1A) and Bip (HSPA5).

It has been recognized that molecular chaperones have a profound effect on the co- and post-translational folding of CFTR at the ER, which could be reflected by changes in the FPMA. Therefore, the proposed siRNA experiments would provide additional insights in the global role of chaperones in both the biogenesis and the peripheral conformational maintenance of CFTR variants. These siRNA studies, however, are compounded by the secondary changes in the expression/activity of multiple members of the chaperone networks as the consequence of chronic downregulation (or inhibition) of the primary target chaperone. This may hamper the interpretation of the results. Specific examples:

a) As requested, we have performed the siRNA-mediated Hsc70 downregulation in CFBE cells, which significantly upregulated Hsp70 (HSPA1A), Hsp90 α , Aha1, and Bip. The expression level of DNAJA1 and DNAJA2 were not affected. These novel results have been presented in Fig.S1A.

b) Pharmacological inhibition of Hsp90 after a few hours can provoke a cellular heat shock response, consisting of the upregulation of e.g. Hsp70 and Aha1, but not DNAJA2 (*Mol. Cell. Proteomics* 2012 11(3):M111.014654 PMID: 22167270).

c) Inhibition of Hsp90 for 24 hours decreased the mature WT-CFTR expression by ~60-70% in CFBE cells as previously reported in BHK cells (*EMBO J* 1998 17: 6879–6887). Cells were treated with the Hsp90 inhibitors Geldanamycin (5 μ M) or Genetespib (100 nM) and the expression of the mature, complex-glycosylated CFTR was measured by immunoblotting

(revised Fig. S1D). Significant alterations in the complex-glycosylated WT CFTR expression were also documented by the siHsp90 α (Appendix Fig.1).

d) Inhibition of Hsp70/Hsc70 for 24 h with VER-155008 abolished trafficking, while MKT-077 increased band C of WT-CFTR (Appendix Fig.2, P Kim Chiaw et al. manuscript in revision). However, titrating with low doses of VER-155008 gives an increase, followed by a decrease in band C (Appendix Fig.2B). VER-155008 is an ATP competitor (*Cancer Chemother Pharmacol.* 2010 66:535-45), while MKT-077 binds to an ADP-bound transient state of Hsp70/Hsc70 (*J. Mol. Biol.* 2011 411:614-32).

Jointly, these observations imply that chronic siRNA exposure induced chaperone depletion may interfere with the ER folding and processing of CFTR with plausible consequences in the channel conformation at the PM. Since our primary goal was to uncover the profolding function of molecular chaperones on the mutant CFTR at the PM (and post-Golgi compartments), we prefer the use of acute inhibition of molecular chaperones in order to minimize secondary effects of siRNAs on the cellular chaperone network activity and rule out of CFTR conformational modulation during its biogenesis at the ER. We are convinced that acute pharmacological inhibition of chaperones has reduced non-specific effects on the cellular chaperone systems and this approach better complements our in vitro studies of Δ F508-CFTR.

3. This work represents a tour-de-force in experimental design and implementation, yet some key links between the experimental paradigms (HeLa cells, BLMs) remain unresolved – sometimes giving a patchwork feeling to the inquiries. As mentioned above, what is the effect of Aha1 knockdown and simultaneous Hsp90/Aha1 knockdown in the HeLa cells (Fig 1)? Effects of DnaJ1, DnaJA2 knockdown? Effects of Hsc70 and Hsp90 inhibitors (and combinations) in the BLMs (Fig 2)? Many of the chaperone effects, especially individually, are somewhat subtle (statistically significant, but with activity well below WT), so it seems imperative to make sure that the effects are consistent across the different experimental paradigms.

1) As requested, we have performed CFTR PM density and functional measurements following the ablation of Hsp90 α , Hsp90 β , Hsp70, Aha1, DNAJA1 and DNAJA2. For these studies, we used CFBE cells, since these are physiologically more relevant models than HeLa cells. Chronic downregulation of these chaperones and co-chaperones significantly increased the PM density and decreased the FPMA of the mutant CFTR after 4 days of treatment. The FPMA was reduced by ~10-35% relative to the NT siRNA, suggesting that chaperones either directly or indirectly contribute to the functional conformation of the mutant. These results are illustrated in Fig.2B and Fig.S1B-C. These experiments, however, are unable to distinguish whether the chaperones-induced conformational modulation is exerted at the ER, post-ER compartment or both locations.

We believe that the simplest and cleanest control experiments are not adding the chaperone as compared to adding chaperone with inhibitor. The former approach has been included (Fig.3A-C, F and I, Fig.S3E and H). Additional control studies have been included to demonstrate the chaperone activity requirement for Δ F508 stabilization as follows.

- A) The contamination level of the isolated microsomes with the most abundant cytosolic chaperones Hsc70 is negligible in the bilayer (Fig.S3B).
- B) Using the mutant variant of Hsc70 with inhibited ATPase activity (Hsc70-K71M) significantly reduced the Hsc70 stabilizing effect of $\Delta F508$ -CFTR in the BLM (Fig. 3I and S4D).
- C) Similar reduction in the thermal stabilization of the channel was observed by omitting the DNAJA2, in the presence of Hsc70/DNAJA1, Hsc70 or DNAJA2 alone (Fig.3I, S4A-C).
- D) Omitting the nucleotide exchange factor GrpE was sufficient to completely curtail the refolding activity of the DnaK/DnaJ complex in the BLM. These new results are present in Fig.4A.
- E) We tested other Hsp70/Hsc70 and Hsp90 inhibitors (MKT-077 and Genetespiib, respectively) in our in vivo studies. Incubation with these drugs elicited similar reduction of the FMPA than Apoptozol + GA acutely in CFBE cells. Using three and two structurally different inhibitors of Hsc70/Hsp70 and Hsp90, respectively, makes it very unlikely that all of them have the same non-specific molecular target with the capacity to stabilize the CFTR. These new data are included as Fig.S1E-F.

In light of the technical difficulties inherent to the $\Delta F508$ CFTR bilayer studies, we feel that performing the requested chaperone inhibition experiment in the BLM, would not provide novel information.

4. Hsc70 binds tightly to lipids and is known to stabilize lysosomes (Arispe 2002 Cell Stress Chaperones 7:330). It isn't clear how much of the effect of Hsc70 on $\Delta F508$ CFTR might arise from direct interactions with the channel and how much is due to important, but indirect, effects on lipid dynamics. Either outcome is interesting, but it seems that the current work would benefit from some clarity on this issue. One way to test it might be to add peptide, NRLLLTG, a well-known competitive ligand of client binding (Zhang et al. 2014 PloS One 9:e103518) that doesn't interfere with lipid binding.

If the mutant channel was stabilized via Hsc70 lipid binding, as suggested, we predict that chaperone mediated stabilization of the channel would be preserved upon inhibiting the Hsc70 substrate binding. To this end, we used the NRLLLTG peptide as a competitive inhibitor of Hsc70-channel substrate binding. In the presence of ATP, however, Hsc70 had a very weak binding to biotinylated NRLLLTG peptide (Appendix Fig.3). The weak association of NRLLLTG with Hsc70 could not be displaced by the free NRLLLTG peptide, confirming the largely non-specific, residual binding of the peptide (Appendix Fig.3). This could be explained by compromised binding of the NRLLLTG peptide to Hsc70 in the presence of mM concentration of ATP, as compared to ADP or in the absence of nucleotides as reported previously (*Biochemistry* 1996, 35, 4636-4644; *J. Biol. Chem.* 2001, 276, 27231-27236). Since ATP omission from the BLM would a) inhibit the cAMP-dependent protein kinase mediated CFTR phosphorylation and cause inactivation of the channel, and b) likely destabilize the conformation of the NBD1-2 sandwich dimer as has been shown for the NBD1 (*Cell* 2012, 148:150-63), we were unable to test the hypothesis that Hsc70 stabilizes the mutant via influencing the BLM dynamics directly.

The following considerations argue for the role of chaperones acting via stabilizing the mutant channel directly in the BLM, rather than indirectly influencing the lipid dynamics.

a) Lipid stabilizing effect of neither the Hsp90 nor the DnaK/DnaJ chaperone system has been reported according to our knowledge, while they have comparable profolding effect on $\Delta F508$ -CFTR as the Hsc70/DNAJA2 in the BLM.

b) Hsp70 localization to the lysosomes and plasma membrane is probably mediated by its high-affinity interaction with lysobisphosphatidic acid (LBPA) and bis(monoacylglycero)-phosphate (BMP) components of the membrane (*J. Exp. Med.* 2004, 200:425–435, *J. Biol. Chem.* 2014 289: 27432–27443). Considering that the BLM does not contain significant amount of either of these lipids, it is unlikely that Hsc70 acts via specific lipid binding.

Minor:

5. DnaJA1 and DnaJA2 are known to have different effects on luciferase refolding (Tzankov 2008 JBC 283:27100, Rauch et al 2014 JBC 289:1402), with DnaJA1 being inactive. This is consistent with the author's observations on the failure of DnaJA1 to restore CFTR in the BLM studies.

Yes, these previous results are consistent with our observation.

6. The experiments with DnaK, DnaJ and GrpE (pg 12-13) are nice, but it isn't clear if they add anything to the narrative.

We feel that the following considerations favor the inclusion of the DnaK results in our manuscript.

a) No previous reports are available documenting the effect of the DnaK/DnaJ/GrpE on the (un)folding energetics of a mutant polypeptide at the single molecule level.

b) We wished to assess whether the extensively studied and evolutionary distant DnaK/DnaJ/GrpE prokaryotic chaperone machinery already possesses the capacity to reshape the unfolding landscape of the mutant CFTR channel, considering that the DnaK, Hsc70 and Hsp90 peptide recognition sites on CFTR are partially overlapping (our unpublished preliminary data).

c) Finally and most importantly, in response to one of the comments of Reviewer#2, the DnaK/DnaJ/GrpE chaperone machine was instrumental to demonstrate the critical role of its ATP turnover driven substrate cycle in the stabilization of $\Delta F508$ -CFTR. It has been documented that GrpE elimination from the DnaK/DnaJ/GrpE complex significantly slows down nucleotide exchange and the refolding of denatured *F. luciferase* (*FEBS Lett.* 1995 368:435-40), which was confirmed in Fig.S4H. Our new results show that the folding activity of DnaK/DnaJ on the mutant channel is virtually abolished upon elimination of GrpE from the complex (see Fig.4A). This results strongly suggests that DnaK undergoes a perpetual binding to and dissociation from $\Delta F508$ -CFTR, which is a prerequisite to render the mutant channel conformationally stable during thermal unfolding by modulating its late-stage (re)folding trajectory.

Reviewer #2

The authors study the impact of chaperones on the folding energetic landscape of $\Delta F508$ CFTR. They investigated the effect of chaperone activity on the functional and conformational maintenance of $\Delta F508$ -CFTR at the PM and following its reconstitution in phospholipid bilayer. They found that inactivation of the temperature-rescued, native-like $\Delta F508$ -CFTR upon thermal unfolding was partially suppressed by Hsc70- and Hsp90-cochaperone both in vivo and in vitro. This was accomplished by kinetic and thermodynamic remodeling of the gating energy landscape of single molecules toward the wild-type channel, an effect mimicked by second-site suppressor mutations. They conclude that chaperones contribute to functional adaptability of $\Delta F508$ -CFTR by reshaping the final fold conformational energetic, a mechanism with implications in influencing (re)folding of ABC-transporters and other PM proteins in health and diseases.

We thank Reviewer #2 his/her detailed critique and constructive suggestions.

Major criticisms

1. Although the authors should be complemented on a large body of work, the manuscript is so densely written and filled with jargon that it will be difficult to have an impact on a general audience.

We have performed an extensive rewrite to make the manuscript more accessible to the general audience.

2. The authors argue that they can address a question of folding by evaluating the effect of unfolding, ie thermal stability. Yet there is no experimental evidence to support this claim.

The reviewer raises a very important point, which we have touched upon in the introduction of the original submission (p.5).

“We chose to investigate the chaperone effect on the channel unfolding rather than the (re)folding process to overcome those technical challenges that are inherent to studies of synchronous co- and posttranslational folding of a newly synthesized $\Delta F508$ -CFTR at the ER and the resistance of denatured CFTR domains to refold. Furthermore, the conformation of the unfolding intermediates of rescued $\Delta F508$ -CFTR may partly overlap with the mutant folding intermediates, suggested by overlapping proteolytic digestion pattern of the immature WT CFTR and the unfolded $\Delta F508$ -CFTR, as well as by VX-809 binding to and stabilizing both the folding and unfolding intermediates of $\Delta F508$ -CFTR.”

In the revised version of the manuscript we eliminated our attempts to extrapolate for the role of molecular chaperones function regarding the modulation of the folding energetic landscape during co-translational folding. In the revised version we restricted our discussion to the refolding activity of the molecular chaperone on the mutant CFTR in post-ER compartments and at the PM.

The following considerations suggest that those $\Delta F508$ CFTR molecules that constitutively or upon corrector rescue escape the ER quality control are subjected to refolding, therefore our simplified model has merit in modeling a physiological process.

- i) A fraction of the $\Delta F508$ -CFTR escapes the ER constitutively and accumulate at 3-4% PM density of the WT (*Physiology (Bethesda)* 2014, 29:265-277; *J. Biol. Chem.* 2001, 276:8942-8950; *Nature* 1992, 358:761-764). Considering the ~5-fold accelerated peripheral turnover of the mutant relative to WT, this implies that ~15-20% of newly synthesized $\Delta F508$ -CFTR can escape the ER as compared to 30-60% of the WT CFTR. Thus, a fraction of the newly synthesized mutant can transiently attain a near-native conformation with the assistance of molecular chaperones, permitting its release from the ER QC.
- ii) The residual $\Delta F508$ -CFTR at the PM unfolds significantly faster than the WT, indicated by its protease susceptibility (*Science* 2010, 329:805-810), as well as its functional and biochemical turnover (e.g. *Science Translational Medicine* 2014, 6(246):246ra97).
- iii) Not only the unfolding of rescued $\Delta F508$ CFTR can be partially inhibited in the BLM (*Nature Chem Biology*, 2013, 9:444-54) or in liposomes (*Chem. Biol.* 2014: 21:666-78), but also the folding of newly synthesized F508del can be improved by small molecule correctors (VX-809) and corrector combinations (*Nature Chem. Biology* 2013, 9:444-54). These results suggest that the conformation of the unfolding intermediates partly overlaps with the folding intermediates of the mutant in the ER.

3. The authors begin the results with supplemental data. The authors should put all relevant data in the body of the manuscript.

Thank you for this excellent suggestion. We have moved the biochemical results of chaperone interaction with the unfolded and near-native $\Delta F508$ CFTR from the Supplement into the main Figure set (Fig.1).

4. The authors measure FPMA a complex measurement of channel activity vs. PM density monitored with PM Elisa. How was this term validated? For example, ablation of Hsc70 increase density but less than channel activity leading to a decrease in FPMA. Was Hsc70 ablated as mentioned? The effects of the drugs are very confusing and explained by overlapping roles of 70 and 90. Finally Fig 1C shows nearly identical results between the DMSO and the drug treated groups. A small effect was only noted at high doses.

To validate the FPMA as a relevant readout for CFTR channel activity, we measured the cell surface density and PKA-activated short circuit current (or the halide-sensitive YFP fluorescence quenching) of rescued $\Delta F508$ CFTR. The FPMA was progressively increased from the basal (~0.01), to forskolin stimulation alone (~0.15) or after corrector and gating potentiator treatment to the maximum value (~0.8-1.0), where the fully activated WT FPMA was designated as 1. These new results are included in Fig.2A. Importantly, changes in the mutant FPMA indicate a similar trend as reported for the open probability changes of $\Delta F508$ -CFTR activation in the absence and presence of CFTR modulators, e.g. gating potentiators increased by ~5-fold the P_o of the $\Delta F508$ CFTR from ~0.08 to 0.42 (*PNAS* 2009, 18825–18830).

The perceived ambiguity of the drug studies in HeLa and CFBE cells was reduced by focusing on the CFBE cells in the main Figure set, and moving most of the HeLa cell results to the Supplement due to reasons discussed in our response to Reviewer#1, point 1.

The Hsc70 knockdown efficiency (~80%) by siRNA has been previously validated in *Science*. 2010 329:805-10 and reconfirmed as shown in revised manuscript (Fig. S1.A)

5. Why was the mutant P67L chosen? Seems out of place and the drugs only had a small effect.

P67L is a milder folding and gating mutant than the Δ F508 CFTR, and the global folding defect caused by this N-terminal mutation is efficiently rescued by VX-809. This mutation helps to demonstrate that the refolding activity molecular chaperones is not Δ F508del CFTR specific, since acute inhibition of Hsc70+Hsp90 reduced the FPMA by >20% (Fig.2D-E).

6. The authors demonstrate conformational resistance with VX-809 in Fig 1H. Do the authors know that this Hsc70 inhibitor is still working in the presence of VX-809? It is possible that VX-809 is having a direct effect on Hsc70/Hsp90 and thus the Pif+GA combination would have less of an effect.

VX809 is a relatively specific pharmacological chaperone of Δ F508 CFTR and is unlikely to interfere with Hsp70 inhibitor activity. To address this concern, we examined the in vivo heat-denatured *F. luciferase* refolding in the presence of VX-809 and chaperone inhibitors in HeLa cells. The refolding of luciferase was profoundly inhibited by both GA+Pif (78.9%) and GA+Apo (78.9%) in vivo (Appendix Fig.4). The inhibitory effect of chaperone drugs was preserved in the presence of VX-809 (92.3% and 87.4 %, respectively, Appendix Fig.4). Therefore, the possibility of direct activation of chaperones by VX-809 can be ruled out.

7. The results on thermal inactivation of the open probability are highly confusing. For example, the revertant mutants never reach wt level 2F, even with the addition of Hsc70 or 90 plus the co-chaperones. A) Is there any evidence that protection from thermal inactivation is directly related to conformational remodeling? B) What if the chaperones bind to the NBD domains and increase open probability directly?

A) The most direct experimental evidence demonstrating the role of Hsc70/Hsp90 activity in the conformational remodeling of the Δ F508 CFTR is the documentation of the ~2-fold increased protease susceptibility of the mutant upon acute inhibition of Hsc70/Hsp90 activity as shown in the revised Fig1E. This conclusion is strengthened by additional evidence:

- a) The turnover of the complex-glycosylated Δ F508 CFTR pool is additively accelerated in the presence of acute inhibition of Hsc70 and Hsp90 (Fig.1D, new results).
- b) The proteases resistance of the low temperature rescued Δ F508 CFTR almost reached that of the WT in isolated microsomes, while its resistance was decreased by ~20 fold after exposing the cells to 37°C for 2 h (*Science* 2010, 329:805-810).

c) Thermal unfolding of the $\Delta F508$ CFTR triggers the association of molecular chaperones (Fig.1A-C), which in turn suppresses the consequences of unfolding and partially masks the functional inactivation of the mutant (as shown in the BLM and by FPMA determinations in vivo).

B) We believe that permanent binding of chaperones to NBDs is insufficient to exert a rescue effect on the mutant CFTR function. Stable chaperone binding would hinder the association-dissociation dynamics of the NBD1-NBD2 heterodimer, which is coupled to channel closing-opening. Hsc70/Hsp90 molecular chaperones do not bind to the native (functional) protein according to the accepted chaperone theory, but only to partially unfolded molecules.

Furthermore, the chaperone foldase activity requires the dissociation-association cycles of the client protein. The chaperone ATPase cycle is necessary to refold $\Delta F508$ CFTR as;

a) the ATPase deficient Hsc70-K71M has reduced refolding activity (Fig.3I) and

b) GrpE, a nucleotide exchange factor of DnaK, is indispensable for the conformational maintenance of the $\Delta F508$ CFTR during thermal unfolding (Fig.4A, new results).

8. In Fig 3A why is wt-CFTR so different from $\Delta F508$ in all the experiments. One would predict that they would be closer.

The $\Delta 508$ mutation profoundly disrupts the co-translational folding and domain assembly of CFTR, leading to the conformational defects in the NBD1/NBD2 and MSD1/MSD2 domains (*Mol Biol Cell*. 2009 20:1903-15). This is illustrated by the ~ 1000 -fold increased protease susceptibility of the full-length mutant as compared to that of the WT (*Science* 2010, 329:805-810). The surprise is that molecular chaperones can make the mutant even partially resemble to WT at all, this is against the one-sequence one-structure idea.

Reviewer #3

The manuscript of Bagdany et al. contains an impressive number of experiments with a wide range of biochemical techniques and functional characterizations. They demonstrate in detail that the folding defect (and consequently, the functional defect) of $\Delta F508$ -CFTR can be partially rescued by lowered temperature, chaperones and other stabilizing agents. The core of this work is the detailed characterization of the gating energy landscape and how it is influenced by unfolding of the mutants during temperature ramps and the different methods of rescue. I find this approach very interesting. However, a number of issues with the presentation and analysis of the data left me thoroughly confused. I cannot decide from the manuscript whether the data are sufficient to support the conclusions. I can therefore not recommend the publication at this stage. I have only listed the more serious issues. My comments below are grouped by topic rather than importance.

We are grateful for the very thorough review and constructive criticism by Reviewer #3.

Biochemical experiments

Page 9: “These suppressor mutants conferred wt-like resistance to $\Delta F508$ -CFTR upon inhibition (by Pif+GA) as indicated by the large unaltered FPMA values (Fig. 1F).” Figure 1F does not prove that the inhibitors have no effect since the control is missing.

We apologize for this oversight. In the revised version, the appropriate controls were included (Fig.2F).

cHSF1 increased PM expression in Fig. 7B and decreases it in Fig. 7C. This apparent contradiction is never mentioned in the text.

The vertical axis of the original Fig.7C was mislabeled, which explains the apparent contradiction. The correct label for the y-axis label is $\Delta F508$ CFTR (PM density or function) (revised Fig.8C). Thus, overexpression of cHSF1 leads to inhibition of the $\Delta F508$ CFTR removal from the PM by ~15% (Fig.8B), while it decreases the steady-state PM density by ~20%. The later may be explained by decreased biosynthetic secretion of the mutant in the presence of cHSF1 overexpression. Importantly, the mutant FPMA was increased by ~50% (Fig.8C, right panel), suggesting that cHSF1 overexpression augmented the function of $\Delta F508$ -CFTR channels.

Thermodynamic analysis

What is the estimated systematic error introduced by the assumption of a system with only two states (O-C) for when the system actually has more states (at least two closed and two open, according to the dwell time histograms)?

We agree that two closed states could be identified for both the WT and mutant channel at 24°C and 26°C. The brief closings largely disappeared or cannot be resolved in the BLM at higher temperatures (Fig.S5A), in contrast to the short-lived open state (which could be distinguished from long open states in the entire temperature range) (Fig. S5B). The short closures were only

slightly longer (~10-30 ms) within open burst periods than those closing events with <10 ms that were routinely excluded from our analysis. (*PNAS* 2010, 107:1241-1246; *J. Gen. Phys.* 2013, 142:61).

To estimate the error introduced by the three-state model, we have recalculated the gating thermodynamics by considering both the short and long closings, which were only detectable at 24°C and 26°C. We were able to fit the short closed dwell time distribution of $\Delta F508$ -CFTR at 24°C and 26°C and of the WT at 24°C. Reliable fitting for two closed states was not possible for the $\Delta F508$ -CFTR activity in the presence of Hsc70+DNAJA2 at 24°C and 26°C. Therefore, the closed states were combined to calculate their weighted mean times (at 24°C and 26°C). Accordingly, the slope of the Arrhenius plot for the $\Delta F508$ -CFTR reduced in the low temperature range (24-30°C), but was unaffected between 30-36°C (Appendix Fig.5A). This manifest in ~4 kJ/mol stabilization of the $\Delta F508$ -CFTR open state at 24°C ($\Delta G_{O-C} = -2.6$ instead of 1.4 kJ/mol). No difference could be resolved in the ΔG_{O-C} value at the physiological temperature (36°C), implying that the three-state model underestimates the open state stability only at low temperature. Correction of the WT mean closed time at 24°C did not cause significant change in slope of the Arrhenius plot (Appendix Fig.5B).

Page 12/figure 3B/C: The change in ΔG when adding chaperones to $\Delta F508$ -2RK in planar lipid bilayers is attributed solely to the change in $T\Delta S$. However, ΔH undergoes about the same dramatic change, both parameters switch from positive to negative, but none of this is mentioned in the text. Does the channel switch between enthalpic and entropic stabilization under different conditions?

As we indicate in the text, the mutant O1_{36°C} and O2_{36°C} states were stabilized by $\Delta G_{O-C} \sim 4.8$ kJ/mol and ~ 3.2 kJ/mol, respectively (Fig. 6A-B and Table S4), partly due to the larger reduction of entropic energy requirement ($T\Delta S_{O-C}$) of opening (from -269 to -27 and from -232 to 4.8 kJ/mol/, respectively) than the simultaneous enthalpy gain at 36°C. This is consistent with the significant chaperone-induced reduction of the conformational disorder of $\Delta F508$ -CFTR. Based on the $\Delta G_{O-C} = \Delta H - T\Delta S$ equation, the ΔH_{O-C} for the opening increased from -258.6 to -21.3 kJ/mol (O1) and from -230.5 to 3.3 kJ/mol (O2) at 36°C in the presence of Hsc70, signifying energetically more favorable open state(s), which is entropy driven in the presence of chaperones. The discussion of these energetic changes at the single molecule level was included on (p.12).

Dwell time histograms

The legends of Fig. 4B/S4A state that you fitted the dwell time histograms with Gaussian distributions instead of exponentials as it is done by virtually every other publication on single-channel recording. If that is true and not a typing error, this decision needs to be explained.

In most cases the event number was too low to permit reliable, multicomponent exponential fittings. Therefore, Gaussian fitting with nonlinear least-squares Levenberg-Marquardt algorithm provided better assessment for the open and closed dwell times. The methodology description has been included in the Method section of the revised manuscript (Supplement p.3).

However, are Fig. 4B/S4A actually standard log-binned dwell time histograms? I am not sure.

The text on page 13 3rd sentence after the paragraph heading speaks of dwell time histograms. The figure legends speak of “cumulative open/closed time histograms” instead. Dwell time histograms and cumulative dwell time histograms are two different things. And since cumulative dwell time histograms are to my knowledge not often used in single-channel analysis, they should be explained a bit.

We apologize for the incorrect labeling of the Figure legend. As stated in the revised text (Fig.5B and Fig.S5A), the results were plotted as the single-channel open and closed log-binned dwell time histograms and obtained by using the Clampfit 10.3 software. The figure legend has been corrected accordingly.

Do Fig. S4A and 4B display the same type of histogram? One has log(open time) on the x-axis, the other log(mean closed time). The latter is highly unusual, please explain.

The correct labeling for both x-axis of the revised Fig.5B and Fig. S5A display the same type of histogram, with log(open time) and log(closed time).

We have corrected the terminology/labeling mistakes and expanded the incomplete description of the single channel analysis in the Methods.

Discussion

On the bottom of page 20, the authors mention the “folding landscape of gating”. Do they suggest that opening and closing of the channel are large-scale folding events? This would be quite provocative and needs a thorough explanation, instead of just being implicated in passing.

According to the dogma, the mutant CFTR should have a single native-like state that is poorly functional. Based on our results, molecular chaperones do more than modulating the poorly folded functional state, they allow a different functional state to be reached most likely by forcing the channel conformational changes along alternative folding pathways.

Based on the energetic analysis of the mutant gating cycle, we show that both the activation energies of gating and the free energy of one closed and two open states of the mutant are altered in the presence of molecular chaperones. These results suggest that the final fold of the closed and open states are indeed altered in the presence of chaperones as a consequence of the modifications of the late-stage of the channel folding pathway. To clarify the terminology, we use “gating energetics” in the revised discussion (p.12).

Presentation

The way the article is written made it difficult for me to read. The main problem I had were not the numerous grammar errors, those could easily be fixed. But some sentences were worded in such a way that I had genuine difficulties understanding them, I suspect that some are even wrong. Other things are downright sloppy, like wrong figure descriptions and a paragraph heading that randomly repeats in the middle of the text. In summary, the

manuscript left me with the impression that it was not proofread by the authors.

We apologize for the indicated deficiencies, which were rectified in the revised version.

I'm listing some of the more confusing examples below:

- Introduction, page 3, 1st paragraph : "... bind to partially folded intermediates and account for maintaining their active conformation." Do the authors imply that those intermediates already active or should that read "reach" rather than "maintain"?

We agree with Reviewer #3, the correct description is: *...account for helping to reach their active conformation.*

- Introduction, page 3, 1st paragraph: That the buffering capacity of chaperones with regards of folding proteins with random mutations enhances genetic diversity is plausible. But how does buffering for environmental insults and errors in protein synthesis increase the genetic diversity?

We have rephrased this sentence in the Introduction (p.3). The revised version reads:

"...It was proposed that the buffering capacity of chaperones could enhance genetic diversity⁵. In response to proteotoxic stresses, chaperones may mask deleterious changes in the folding energy landscape^{3,6}, which can be beneficial in a changing environment."

- The sentence between pages 3 and 4 contains no less than 70 words.

We have rephrased the message and split the sentence.

- Results, page 5, the long sentence at the end of the 2nd paragraph. The digestion patterns of partially folded wt and unfolded mutant are similar. The authors conclude from this that it is plausible that the intermediates of the mutant during folding and refolding are similar to each other. I don't understand this conclusion.

In the revised version we have largely restricted our discussion to the refolding/conformational stabilization of the near-native $\Delta F508$ -CFTR at the PM. Therefore, we omitted this ambiguous conclusion from the resubmission (p.17, first paragraph).

- Legend to Figure S1A: "...was either unfolded for 2.5h at 37° (the main text referring to this figure on page 6 says 2 hours) in the presence of ... (lanes 3-6) or exposed to 37° for 20 min and then cultured at 26° for 12 hours". Where is this second condition shown and why was it done?

We apologize for omitting the rationale due to space limitation. The revised version includes an explanation as follows:

“.....Then, to preserve the near-native conformation or unfold the rescued $\Delta F508$ -CFTR, cycloheximide (CHX) chase was performed at 26°C for 10 h or at 37°C for 2 h, respectively. The CHX-chase was necessary to eliminate partially folded core-glycosylated $\Delta F508$ and WT CFTR forms the ER (band B, revised Fig.1A), representing folding intermediates prone to chaperone association” (p.6). The short, 20 min 37°C incubation was required to facilitate the elimination of the core-glycosylated form, but minimize unfolding and it was established in our pilot studies.

- Legend to Figure S2B: “Association of Hsc70, but not Hsp90 with CFTR was dependent on the ATP concentration.” I don’t understand this statement. The dark grey bars with ATP do not differ substantially from the red bars without ATP. The association seems to depend on “Apy” instead.

Consistent with ATP-dependent substrate binding, ATP-depletion by Apyrase (Apy) augmented the association of Hsc70 and the mutant channel (Fig. 1B). The revised text was corrected accordingly.

- Figure 1A has “FPMA” on the y-axis, but displays, per the legend the PM density and the I-efflux individually, not their ratio.

The Y-axis labeling has been corrected to % $\Delta F508$ PM density or function.

- The legend to Figure 1A,B states 2.5 hours of heat stress, the text referring to the figure states 2 hours.

We usually used 2-2.5 hours of heat stress depending on the cellular model. For HeLa cells we used routinely 2.5 hours of heat stress and the text was corrected accordingly.

- The empty arrowhead in Fig. 1C is only explained somewhere in the Supplements.

We introduced the explanation of the empty arrowhead (core-glycosylated form) in the legend of Fig.1A.

- Page 8, referring to Fig. 1F: “Inhibition of Hsc70/Hsp70 and Hsp90 in combination with Pif+GA ...”. There is no hint in the figure of what additional chaperone inhibition was combined with Pif and GA. Did the authors perhaps mean “Inhibition by a combination of Pif+GA”?

Yes, the Reviewer is correct, we meant inhibition by a combination of Pif+GA. We corrected the text accordingly.

- Page 10, first sentence: “We reconstituted the mutant inactivation process in BLM.” How does one reconstitute a process? Should the sentence read “We reconstituted the mutant into BLMs to observe the inactivation process?”

We corrected the text as suggested.

- The legend to Figure S2D states “WT-CFTR activity was monitored at ~34° for ~6 min”. The figure itself is labelled with a temperature gradient from 27.8° to 35.2° and back to 34.4°. To my estimate, the time spent between 34° and 35° could be what the authors meant in the legend. Please clarify.

The legend to the revised Fig.S3a was corrected as “After a temperature ramp, WT-CFTR activity was monitored for an additional ~6 min at 34-35°C.”

- Page 12, last paragraph. What is “unfolding gating kinetics”? Do you mean unfolding kinetics or gating kinetics?

The Reviewer is correct, and we changed the text accordingly.

- Materials and Methods: the paragraph heading “Reconstruction of CFTR...” is repeated in the middle of the text for no apparent reason.

The repetition was due to an editing mistake, we apologize. The text was corrected accordingly.

[Appendix Figures redacted]

REVIEWERS' COMMENTS:

Reviewer #1 (Remarks to the Author):

The revised manuscript is significantly improved. The role of Hsp90 is more clear and there is better coherence amongst the experimental systems.

Reviewer #2 (Remarks to the Author):

The authors should be commended for their revisions to his manuscript, which have clearly resolved many of the points raised in the previous work. However, it remains difficult to decipher the message that the authors want to convey. The experimental protocol involving low to high to low temperature and cycloheximide treatment is complex. This protocol may have scientific value but the relevance for CF is quite low. Current thinking is that the thermal instability in $\Delta F508$ -CFTR is present at translation. $\Delta F508$ -CFTR never gets to a stable state as achieved at low temperature. Again raising the question of relevance. Fig 1 shows that there is more binding at 37°C, which makes sense, but the binding is unproductive in that the $\Delta F508$ -CFTR unfolds at this temperature and thus the chaperones cannot prevent unfolding in nature. The title states that: Chaperones Rescue the Energetic Landscape of Mutant CFTR. Actually they don't because they don't rescue $\Delta F508$ -CFTR under normal conditions. It is significant that activating Hsp90 does partially rescue $\Delta F508$ -CFTR. Does blocking chaperone function make it harder to rescue with VX-809 and is VX-809 function augmented by activating Hsp90. The partial rescue of Hsp90 by activating Hsp90 is not novel given that it has already been shown that altering Aha1 can also rescue $\Delta F508$ -CFTR. The authors make the comment in the Discussion, that this could explain variation in phenotype. I think that if the authors could prove that it would point very clearly toward the relevance of their work. Perhaps the relevance of the work is that the chaperones rescue the energetic landscape to stabilize the $\Delta F508$ -CFTR enough so that it can be rescued by VX-809. Finally, all of the work was done in experimental cells. Some use of primary cell lines is required to improve the impact.

Reviewer #3 (Remarks to the Author):

One major cause of cystic fibrosis is the deletion of Phe508 in the CFTR protein, causing a folding defect, more specifically thermal unfolding at 37°C. The work by Bagdani et al. investigates the role of chaperones in the partial rescue of the folding defect. They expressed DeltaF508-CFTR at low temperatures and observed the loss of membrane expression and function upon return to physiological temperatures. Through a large number of different assays, they demonstrate that chaperones, especially HSP70 are able to partially prevent the thermal unfolding and stabilize the channel in a state that is not quite the same as the wildtype, but functionally closer to it both in function and density on the plasma membrane.

This is the revised version of a manuscript that I had reviewed previously. All my previous concerns were addressed (and solved, in most parts) in the rebuttal letter by the authors. In response to the others reviewers, the manuscript has also been strongly rewritten and new experiments were added. The authors have clearly put a lot of work into the revision and I now can follow the general line of the text much more clearly than before.

The work the authors have done is overwhelming in both senses of the word. The good: I can only compliment them on the huge amount of work that they must have put into the experiments to thoroughly shed light on the topic from all angles. The bad: As a reader, who is not intimately familiar with all experimental techniques used, I felt overwhelmed with the huge amount of detail contained in the ~120 individual graphs (not counting dwell time histograms for different temperatures) contained in 16 figures in the manuscript and supplements.

Often I had the impression that the authors were trying to squeeze several topics into one sentence to fit the huge amount of content into the length restrictions of the journal. I suspect that this is the main reason for some of the apparent contradictions that I tripped over also in this second version of the manuscript.

Perhaps it would be a good idea to split the paper into two, one biochemical and one with the bilayer data? This would allow more detailed explanations and increase accessibility. The results of this work are interesting and provocative, and it would be a shame to bury them in a paper that is not accessible to a wide audience.

I have listed below a few instances where I was able to pinpoint what exactly was confusing me.

1) PM density/macroscopic function assay

1A)

Fig. 2A and Text on p. 7, bottom

„The FPMA was validated by demonstrating its profound increase from 0.2 to 0.8-1 upon correction of the Δ F508-CFTR processing and gating defects by VX-809 and VX-770, respectively, to a value similar to that of WT in CFBE cells (Fig.2A).“

1B)

While VX-809 indeed increases both the PM density and I_{sc} of the mutant, it clearly decreases the FPMA, in contrast to the text. Furthermore, VX-770 is never mentioned in Fig. 2A. No explanation is offered why the wt was activated by forskolin alone and the mutant by fsk+gen, and how these data are comparable when they were acquired under different conditions. (In Figure 2C/E/G, both frk and gen were used also on the wt. But why not in Fig. 2A?)

1C)

Why is the current of the Δ F508 mutant so much larger in Fig. 2C compared to Fig. 2A, even though the enhancer VX-809 was not used in C?

1D)

Legend of Fig. 2B. „Cells without Δ F508 + NT siRNA served as controls“ for the effect of siRNA. Would it not be better to use cells with the channel, to rule out an effect of NT siRNA?

Legend of Fig. 2F mentions P67L, the figure does not.

2) BLM recordings

2A)

The open probability of Δ F508-CFTR-2RK in Fig. 3C seems to increase with temperature. This does not quite match the plots in Figs. 3F-H and the text. (If one would include the open-channel flicker of the channel at higher temperatures, the p_o might actually decrease as in Figs. 3F-H. But this would have to be explained briefly. As it is now, the comparison of Fig. 3C and F-H is just confusing)

2B)

Figure 3 and the related text on Page 11 left me confused where the Δ F or the Δ F-RK mutant was meant.

Are all the labels and figure legends in Fig. 3 correct regarding whether Δ F508 or Δ F508-RK were used? There is a sentence on p. 11, that suggests that the same mutants were used in Fig. 3D, F and I:

„In the presence of Hsc70/DNAJA2 the p_o increased by ~2-fold at 36°C (Fig. 3D, F and I).“

However, in the figure, F and I contain only the wt and Δ F-RK, while D is labelled Δ F, without RK.

(Even though the two variants behave similar according to the literature, it should be clear what is displayed in which figure and which mutant is meant in the text.)

2C)

Supplemental Figure S4, please check the figure labels a-f and the legend regarding which chaperones were added. Several of them do not match. In f, I don't see that pO has recovered upon return to 25°. In fact, in the last half of the lowermost panel, the open probability is the lowest of the whole trace.

2D)

The common definition for the term "cumulative histogram" is "a mapping that counts the cumulative number of observations in all of the bins up to the specified bin." (I admit, I copied that from Wikipedia). The histograms you show in Figs. 5 and S5 are clearly standard, log-binned dwell time histograms, not cumulative. So I must repeat my question from the first review: What do you mean when you use "cumulative dwell time histogram" in the figure legends? Did you perhaps pool the data from different records?

3) Supplement - Single-channel analysis

3A)

Supplement, page 3:

"Single channel amplitudes were determined from amplitude histograms. We used Gaussian distribution with nonlinear least-squares Levenberg-Marquardt algorithm for histogram fitting, since this method resulted in smaller error than exponential fits especially in case of mutants."

3B)

Fitting amplitude histograms with exponentials makes no sense anyway. Are you by chance mixing in this sentence the determination of single-channel current with the dwell time histogram fits?

4) Supplement - Isc

On p. 5 of the supplement, it is stated that

"CFTR-mediated iodide transport was calculated from peak value of iodide release after normalizing for protein content."

The related figures, 2A, middle panel and 2C show no hint or normalization for protein content. According to the y-axis of the figures, the normalization was done per membrane area. Does the sentence in the appendix perhaps try to describe Isc and the FPMA at the same time?

5) Supplement - DeltaH and TDeltaS

Fig. S5E. None of the blocks of bars is labelled, so I don't know which one represents which channel and/or chaperone. All I can see is that none of them matches the wt CFTR in Fig. S7H, despite the legend mentioning that the wt is included in 5E as a control.

ANSWERS FOR REVIEWERS' COMMENTS:

Reviewer #1:

The revised manuscript is significantly improved. The role of Hsp90 is more clear and there is better coherence amongst the experimental systems.

We thank the reviewer for appraising the revised version of our manuscript.

Reviewer #2:

The authors should be commended for their revisions to his manuscript, which have clearly resolved many of the points raised in the previous work. However, it remains difficult to decipher the message that the authors want to convey. The experimental protocol involving low to high to low temperature and cycloheximide treatment is complex. This protocol may have scientific value but the relevance for CF is quite low. Current thinking is that the thermal instability in $\Delta F508$ -CFTR is present at translation. $\Delta F508$ -CFTR never gets to a stable state as achieved at low temperature. Again raising the question of relevance.

Thank you for recognizing the improvement in our manuscript. We have tried to address all the points raised during the revision.

As we have shown previously, $\Delta F508$ -CFTR has both temperature-sensitive folding and stability defect (e.g. J Biol Chem. 2001 276:8942-50). The low temperature protocol serves as a tool to accumulate sufficient amount of mutant at the PM that is amenable to biochemical and functional studies without introducing an artifact. This latter conclusion is underlined by the fact that a residual amount of $\Delta F508$ -CFTR is documented at the cell surface in primary respiratory and intestinal epithelia, which reaches ~3-5% that of the WT-CFTR (e.g. Van Goor, PNAS, 2011, 18843–18848, and see p.5). Based on the 4-fold faster turnover of the $\Delta F508$ -CFTR in post-ER compartments, one can infer that ~12-20% of newly synthesized mutant can constitutively escape the ER quality control relative to that of WT-CFTR, which is not an insignificant amount. Therefore, a fraction of the mutant attains, at least temporarily, near-native conformation and escapes the ER quality control under physiological conditions. In post-ER compartments, however, these conformers partially unfold and are recognized by the quality control - depending on the recognition capacity and activity of cellular chaperone systems - and subjected to refolding and/or degradation. Thus, the ultimate fate of the mutant will be dependent on both profolding and prodegradative activity of the proteostasis system, as well as the severity of the folding defect. While low temperature rescue was not necessary for the sizable accumulation of a less severe folding mutation (e.g. P67L), this mutation is still susceptible for chaperone-mediated refolding (see Fig. 2E). Thus understanding and modulating the fate of $\Delta F508$ -CFTR and other folding mutants with incomplete ER retention is highly relevant from both basic and translational research.

The strength of our paper is that it demonstrates the constitutive profolding activity of the two major chaperone systems and their role in partially suppressing the loss-of-function phenotype both at the biochemical and transport levels in two different cellular models, using two CF causing mutants, as well as at the single molecule level after reconstitution of the channel into BLM. In addition, we show that Hsp90 activation by small molecules can partially rescue the functional stability defect of the $\Delta F508$ -CFTR (Fig.8 D-E), an approach that may be utilized in future therapeutic attempts.

Fig 1 shows that there is more binding at 37oC, which makes sense, but the binding is unproductive in that the $\Delta F508$ -CFTR unfolds at this temperature and thus the chaperones cannot prevent unfolding in nature. The title states that: Chaperones Rescue the Energetic Landscape of Mutant CFTR. Actually they don't because they don't rescue $\Delta F508$ -CFTR under normal conditions.

We respectfully disagree with the interpretation of the reviewer. We demonstrate that the cytosolic chaperone activity can constitutively rescue the mutant unfolding in post-ER compartment both at the biochemical and functional level in vivo. Without this process the mutant expression would be further attenuated (see Fig. 2B-C,G and Fig.8). The definitive proof for the chaperone profolding activity on the mutant is demonstrated by the bilayer experiments using three different chaperone systems. The documented change in the mutant gating energetics toward the WT channel by chaperones yields further evidence that chaperone systems can modulate the final fold conformation at the single molecule level. Please, note that we avoid stating that molecular chaperones can fully rescue the mutant folding defect in the entire manuscript, since none of our data support this inference.

It is significant that activating Hsp90 does partially rescue $\Delta F508$ -CFTR. Does blocking chaperone function make it harder to rescue with VX-809 and is VX-809 function augmented by activating Hsp90. The partial rescue of Hsp90 by activating Hsp90 is not novel given that it has already been shown that altering Aha1 can also rescue $\Delta F508$ -CFTR.

Previous publication has shown that inhibiting Hsp90 activity suppressed WT CFTR folding (EMBO J. 1998, 17(23):6879-87), an observation that we have confirmed. Therefore, the proposed experiments are technically not feasible. Chronic downregulation of Aha1 (Cell 2006, 127, 803–815), which partially inhibits the Hsp90 ATPase activity, indeed diminished the FPMA of the mutant (Fig.2B). On the other hand, the partial rescue of the mutant by allosteric activation of Hsp90 is a novel observation. Balch and co-workers used siRNA to ablate Aha1 biochemical expression (and function), which leads to increased expression of the mutant, but no detailed cellular mechanism has been provided. We have shown that Aha1 ablation stabilize the mutant CFTR at the PM (Science 329, 805 (2010)). In the present work we monitored the mutant responsiveness selectively at the cell surface upon acute activation of Hsp90 with small molecules, isolating the cell surface effect of Hsp90 activation from its impact on the ER processing. Another notable difference is that the previous study by Balch and co-workers did not determine the cell surface channel activity and density simultaneously, therefore no inference could be made for the functional state at the single channel level.

The authors make the comment in the Discussion, that this could explain variation in phenotype.

I think that if the authors could prove that it would point very clearly toward the relevance of their work. Perhaps the relevance of the work is that the chaperones rescue the energetic landscape to stabilize the $\Delta F508$ -CFTR enough so that it can be rescued by VX-809. Finally, all of the work was done in experimental cells. Some use of primary cell lines is required to improve the impact.

We fully agree with the reviewer that this would be an interesting and important aspect of our work. However, since the manuscript already contains overwhelming information, as pointed out by reviewer #3, the proposed experiments appear to be beyond the scope of our manuscript and will be pursued in a follow up study.

Reviewer #3

One major cause of cystic fibrosis is the deletion of Phe508 in the CFTR protein, causing a folding defect, more specifically thermal unfolding at 37°C. The work by Bagdani et al. investigates the role of chaperones in the partial rescue of the folding defect. They expressed $\Delta F508$ -CFTR at low temperatures and observed the loss of membrane expression and function upon return to physiological temperatures. Through a large number of different assays, they demonstrate that chaperones, especially HSP70 are able to partially prevent the thermal unfolding and stabilize the channel in a state that is not quite the same as the wildtype, but functionally closer to it both in function and density on the plasma membrane.

This is the revised version of a manuscript that I had reviewed previously. All my previous concerns were addressed (and solved, in most parts) in the rebuttal letter by the authors. In response to the others reviewers, the manuscript has also been strongly rewritten and new experiments were added. The authors have clearly put a lot of work into the revision and I now can follow the general line of the text much more clearly than before.

The work the authors have done is overwhelming in both senses of the word. The good: I can only compliment them on the huge amount of work that they must have put into the experiments to thoroughly shed light on the topic from all angles. The bad: As a reader, who is not intimately familiar with all experimental techniques used, I felt overwhelmed with the huge amount of detail contained in the ~120 individual graphs (not counting dwell time histograms for different temperatures) contained in 16 figures in the manuscript and supplements.

Often I had the impression that the authors were trying to squeeze several topics into one sentence to fit the huge amount of content into the length restrictions of the journal. I suspect that this is the main reason for some of the apparent contradictions that I tripped over also in this second version of the manuscript.

Perhaps it would be a good idea to split the paper into two, one biochemical and one with the bilayer data? This would allow more detailed explanations and increase accessibility. The results of this work are interesting and provocative, and it would be a shame to bury them in a

paper that is not accessible to a wide audience.

I have listed below a few instances where I was able pinpoint what exactly was confusing me.

We are indebted for the valuable suggestions and critiques that significantly improved our manuscript.

1) PM density/macrosopic function assay

1A) *Fig. 2A and Text on p. 7 , bottom*

„The FPMA was validated by demonstrating its profound increase from 0.2 to 0.8-1 upon correction of the $\Delta F508$ -CFTR processing and gating defects by VX-809 and VX-770, respectively, to a value similar to that of WT in CFBE cells (Fig.2A).“

1B) While VX-809 indeed increases both the PM density and I_{sc} of the mutant, it clearly decreases the FPMA, in contrary to the text. Furthermore, VX-770 is never mentioned in Fig. 2A. No explanation is offered why the wt was activated by forskolin alone and the mutant by $fsk+gen$, and how these data are comparable when they were acquired under different conditions. (In Figure 2C/E/G, both frk and gen were used also on the wt. But why not in Fig. 2A?)

The FPMA of the mutant became comparable to that of the WT upon the phosphorylation in the presence of forskolin and activation by genistein, but not upon application of forskolin alone as a consequence of the inherent gating defect of the $\Delta F508$ -CFTR. This observation demonstrates that the FPMA can be used as a surrogate indicator for the open probability of the cell surface resident channel.

The reduction of FPMA of WT CFTR in the presence of VX-809 was small (mean= 0.96 ± 0.13 and 0.79 ± 0.1 respectively, n=3) and not significant (p=0.1).

As pointed out by the reviewer, in Fig.2A we used the gating potentiator genistein and not VX-770. This mistake has been corrected.

Since the WT-CFTR forskolin-stimulated open probability has been well established, it served as the reference point for the FPMA of the mutant. As requested, in the revised version we also indicate the modestly increased activation of WT by genistein (Fig.2A).

1C) *Why is the current of the DeltaF508 mutant so much larger in Fig. 2C compared to Fig. 2A, even though the enhancer VX-809 was not used in C?*

In Fig. 2C the $\Delta F508$ -CFTR was rescued by low temperature (48h, 30°C). This is stated in the legend.

1D) Legend of Fig. 2B. “Cells without DeltaF508 + NT siRNA served as controls” for the effect of siRNA. Would it not be better to use cells with the channel, to rule out an effect of NT siRNA?

The reviewer is correct, the wording was misleading. $\Delta F508$ expressing cells were transfected with NT siRNA. CFTR-deficient cells were used to determine non-specific antibody binding in the PM ELISA assay for the FPMA calculation. The latter results are not illustrated. The legend has been corrected.

Legend of Fig. 2F mentions P67L, the figure does not.

P67L was moved to Fig 2D without updating the figure legend. This mistake was corrected.

2) BLM recordings

2A) The open probability of deltaF508-CFTR-2RK in Fig. 3C seems to increase with temperature. This does not quite match the plots in Figs. 3F-H and the text. (If one would include the open-channel flicker of the channel at higher temperatures, the pO might actually decrease as in Figs. 3F-H. But this would have to be explained briefly. As it is now, the comparison of Fig. 3C and F-H is just confusing)

The majority of mutant channels are highly thermosensitive (Fig. 3C, left panel). A small fraction of the mutant channels was more resistant to thermal inactivation as illustrated on the right panel. The justification of including a less typical record is indicated in the revised legend.

2B) Figure 3 and the related text on Page 11 left me confused where the DeltaF or the DeltaF-RK mutant was meant. Are all the labels and figure legends in Fig. 3 correct regarding whether DeltaF508 or DeltaF508-RK were used? There is a sentence on p. 11, that suggests that the same mutants was used in Fig. 3D, F and I: “In the presence of Hsc70/DNAJA2 the P_o increased by ~2-fold at 36°C (Fig. 3D, F and I).” However, in the figure, F and I contain only the wt and DeltaF-RK, while D is labelled DeltaF, without RK. (Even though the two variants behave similar according to the literature, it should be clear what is displayed in which figure and which mutant is meant in the text.)

The labelling was incorrect indeed; we thank the reviewer for recognizing this mistake. In Fig. 3 we used $\Delta F508$ -2RK in all panels, except in panel B. We corrected the Fig. 3D-E labelling. In panels F and I we do not show $\Delta F508$ without 2RK indeed, but the temperature dependent activity of $\Delta F508$ is shown in Fig S3E as well. The text was changed to reflect the usage of the correct constructs.

2C) Supplemental Figure S4, please check the figure labels a-f and the legend regarding which chaperones were added. Several of them do not match. In f, I don't see that pO has recovered upon return to 25°. In fact, in the last half of the lowermost panel, the open probability is the lowest of the whole trace.

We corrected the panel labels of Fig. S4.

The results of the experiment illustrated on Fig. S4F suggest that the mutant functional stability is reversibly influenced by thermal stress under our experimental condition. The P_o values for the 25-30 °C intervals before and after the temperature ramp were 0.13 and 0.16, respectively, indicating that the channel did not undergo irreversible thermal denaturation.

2D) The common definition for the term “cumulative histogram” is “a mapping that counts the cumulative number of observations in all of the bins up to the specified bin.” (I admit, I copied that from Wikipedia). The histograms you show in Figs. 5 and S5 are clearly standard, log-binned dwell time histograms, not cumulative. So I must repeat my question from the first review: What do you mean when you use “cumulative dwell time histogram” in the figure legends? Did you perhaps pool the data from different records?

The data were pooled from $n= 3-24$ different records. As suggested, in the revised version we use the term dwell time histogram.

3) Supplement - Single-channel analysis

3A) Supplement, page 3: “Single channel amplitudes were determined from amplitude histograms. We used Gaussian distribution with nonlinear least-squares Levenberg-Marquardt algorithm for histogram fitting, since this method resulted in smaller error than exponential fits especially in case of mutants.”

3B) Fitting amplitude histograms with exponentials makes no sense anyway. Are you by chance mixing in this sentence the determination of single-channel current with the dwell time histogram fits?

The methodological description was corrected. We determined the dwell times from Gaussian distribution with nonlinear least-squares Levenberg-Marquardt algorithm. The amplitude histograms were fitted with Gaussian as well.

4) Supplement – Isc On p. 5 of the supplement, it is stated that “CFTR-mediated iodide transport was calculated from peak value of iodide release after normalizing for protein content.” The related figures, 2A, middle panel and 2C show no hint or normalization for protein content. According to the y-axis of the figures, the normalization was done per membrane area. Does the sentence in the appendix perhaps try to describe Isc and the FPMA at the same time?

The sentence “CFTR-mediated iodide transport was calculated from peak value of iodide release after normalizing for protein content.”, referred to the functional measurements in HeLa cells (FigS. 1G). Fig. 2A and 2C depict the channel function in CFBE monolayers, determined by short circuit current measurement (Isc). Isc was normalized for the monolayer area ($\mu\text{A}/\text{cm}^2$) in CFBE cells (FigS. 2A middle panel and 2C). The figure description was corrected accordingly.

5) Supplement – ΔH and $T\Delta S$

Fig. S5E. None of the blocks of bars is labelled, so I don't know which one represents which channel and/or chaperone. All I can see is that none of them matches the wt CFTR in Fig. S7H, despite the legend mentioning that the wt is included in 5E as a control.

We included the missing labels in Fig. S5E. In Fig. S5E and Fig. S7H, we show different mutants ($\Delta F508$ - with 2RK and solubilisation mutations respectively). We used only single channel records for analysis in Fig. S5E, while in Fig. S7H both single and multiple channels, which may provide a plausible explanation for the observed difference in the WT values. While the absolute numbers of the gating thermodynamics differ to some extent, the tendency of changes between WT and mutant and the chaperone effect are similar. Chaperones can partially restore the ΔH_{O-C} and $T\Delta S_{O-C}$ values of $\Delta F508$ -2RK towards that of the WT, similar to $\Delta F508$ -2RK containing R1070W or R1070W/3S, but not the 3S suppressor mutations.